# Blood genome expression profiles in infants with congenital cytomegalovirus infection

Christopher P. Ouellette [1], Pablo J. Sánchez[1,2,3,4], Zhaohui Xu [5], Derek Blankenship[6], Fiker Zeray[7], Andrea Ronchi [4,11], Masako Shimamura[1,5], Damien Chaussabel [8], Lizette Lee [4,7], Kris E. Owen[7,9], Angela G. Shoup[9,10], Octavio Ramilo [1,5] & Asuncion Mejias [1,5 ✉]

Congenital CMV infection (cCMVi) affects 0.5–1% of all live births worldwide, making it the leading cause of sensorineural hearing loss (SNHL) in childhood. The majority of infants with cCMVi have normal hearing at birth, but are at risk of developing late-onset SNHL. Currently, we lack reliable biomarkers to predict the development of SNHL in these infants. Here, we evaluate blood transcriptional profiles in 80 infants with cCMVi (49 symptomatic, 31 asymptomatic), enrolled in the first 3 weeks of life, and followed for 3 years to assess emergence of late-onset SNHL. The biosignatures of symptomatic and asymptomatic cCMVi are indistinguishable, suggesting that immune responses of infants with asymptomatic and symptomatic cCMVi are not different. Random forest analyses of initial samples in infants with cCMVi, irrespective of their clinical classification, identify a 16-gene classifier signature associated with the development of SNHL with 92% accuracy, suggesting its potential value as a biomarker.

[1] Department of Pediatrics, Division of Pediatric Infectious Diseases, Nationwide Children's Hospital –The Ohio State University College of Medicine, Columbus, OH, USA. [2] Division of Neonatology, Nationwide Children's Hospital–The Ohio State University College of Medicine, Columbus, OH, USA. [3] Center for Perinatal Research, Abigail Wexner Research Institute at Nationwide Children's Hospital, Columbus, OH, USA. [4] Department of Pediatrics, Divisions of Neonatal-Perinatal Medicine and Pediatric Infectious Diseases, University of Texas Southwestern Medical Center, Dallas, TX, USA. [5] Center for Vaccines and Immunity, Abigail Wexner Research Institute at Nationwide Children's Hospital, Columbus, OH, USA. [6] DB Analytics LLC, Dallas, TX, USA. [7] Children's Medical Center Dallas, Dallas, TX, USA. [8] Sidra Medical and Research Center, Doha, Qatar. [9] Department of Otolaryngology, University of Texas Southwestern Medical Center, Dallas, TX, USA. [10] Parkland Health and Hospital System, Dallas, TX, USA. [11] Present address: Neonatal intensive care unit, Fondazione IRCCS, Ca' Granda Ospedale Maggiore Policlinico, 20122 Milan, Italy. ✉email: Asuncion.Mejias@nationwidechildrens.org

Human cytomegalovirus (CMV), otherwise known as human herpesvirus 5, is a *betaherpesvirus* of the order *Herpesviridae*, a group of eight double-stranded DNA viruses that establish lifelong latency after infection[1]. Infection with CMV can produce a wide spectrum of disease, ranging from asymptomatic infection to severe multiorgan systemic disease in the susceptible host. Importantly, in-utero infection of the fetus can result in congenital infection with CMV. Congenital cytomegalovirus (cCMV) infection remains a major global public health problem, affecting 0.5 to 1% of all live births[2]. Fetal infection with CMV results in sensorineural hearing loss (SNHL) in ~33% to 44% of infants with clinically apparent (symptomatic) disease, and in 10% of well-appearing infants with no clinically apparent signs of disease at birth (asymptomatic). As a result, cCMV is the leading cause of non-genetic hearing loss in childhood[3–6]. Importantly, up to 25% of infants with cCMV infection will have normal hearing at birth yet develop hearing loss later in life (late-onset)[2]. In addition, cCMV infection is a major contributor to permanent neurologic disabilities and cognitive deficits in childhood[7].

Despite the long-term impact of cCMV infection, there are no reliable biomarkers that correlate with clinical disease manifestations, or help identifying infants at increased risk for serious clinical outcomes. Prior studies have evaluated the utility of CMV loads in blood or urine as a biomarker predictive of SNHL[8,9]; however, no specific viral loads have shown to correlate with hearing loss in the symptomatic or asymptomatic infant. Analysis of host blood transcriptional profiles has provided significant insights regarding disease pathogenesis, diagnosis, assessment of clinical severity and improved patient classification of children with varied ailments[10–14]. Recently, this approach has been explored in a small cohort of infants with cCMV infection utilizing dried blood spots, identifying preliminary associations of transcriptional signatures with long-term outcomes[15].

By applying whole-blood transcriptomics, we sought to identify the differences in gene expression profiles between infants with symptomatic and asymptomatic cCMV infection and their association with late-onset SNHL. Our data show that blood immune profiles between symptomatic and asymptomatic infants with cCMV infection are indistinguishable, suggesting that host responses to congenital CMV infection are similar irrespective of the clinical classification. In addition, we identify a 16-gene classifier set that distinguishes with 92% accuracy infants who would develop late-onset SNHL, suggesting the value of host genomic responses as a biomarker for hearing loss in congenital CMV infection.

## Results

**Patient characteristics**. Eighty-six infants with cCMV infection and 21 healthy control infants were enrolled. Of those, samples from six cCMV-infected infants and 11 healthy controls were excluded owing to insufficient or low-quality RNA, resulting in a study cohort of 80 infants with cCMV infection and 10 healthy controls (Supplementary Fig. 1). The demographic and clinical characteristics of cCMV patients and healthy controls included and not included in the study were comparable (Supplementary Tables 1 and 2). The baseline demographic characteristics of healthy controls and infants with symptomatic and asymptomatic cCMV infection included in downstream analyses are included in Table 1.

The diagnosis of infants with symptomatic and asymptomatic cCMV infection was established in the first week of life per standard of care, and blood samples were obtained at study enrollment the second or third week of life (median age 17 days). The overall median gestational age was 39 weeks in both cCMV groups and 38 weeks in the healthy control cohort, but there were no significant differences in gestational age between the discovery cCMV cohorts and healthy controls (Supplementary Table 3). Infants with symptomatic cCMV infection had significantly smaller head circumferences, and 16% had microcephaly. Blood polymerase chain reaction (PCR) was performed in 31% (25/80) of patients, and CMV DNA detected in 6 (24%) infants. Only one of those six infants had a quantitative rt-PCR performed (12,668 CMV copies/mL). Overall, 19 (24%) infants (all symptomatic) received antiviral therapy for 6 weeks or 6 months. No infant with asymptomatic cCMV infection had evidence of SNHL at birth, compared with 9 (11%) infants with symptomatic cCMV infection. During the 3-year longitudinal follow-up period, 13 (27%) infants with symptomatic cCMV and 11 (35%) with asymptomatic cCMV infection developed late-onset SNHL. The demographic, clinical, laboratory, neuroimaging findings at diagnosis, and results of audiologic evaluations at diagnosis and follow-up for both cCMV cohorts are summarized in Table 2.

**Transcriptional signatures of cCMV infection**. To define and validate the blood transcriptional biosignatures for infants with symptomatic ($n = 49$) and asymptomatic ($n = 31$) cCMV infection, patient samples in each of these cohorts were divided randomly in two groups: training (discovery) and test set (validation), for a total of four groups. The same 10 healthy control infants who were matched for age, sex, gestational age, and race with the discovery cohorts, were used for comparisons in the validation sets (Supplementary Table 3). Baseline samples for transcriptome analyses were collected before initiation of antiviral therapy with the exception of 2 (3%) patients who were receiving valganciclovir for 3 and 6 days, respectively, at first study visit. Follow-up samples on year 1, 2, and 3 were collected off antiviral therapy.

The symptomatic cCMV biosignature was derived in the discovery cohort (training set; $n = 25$) and validated in a second independent group of symptomatic cCMV-infected infants (test set, $n = 24$). Statistical group comparisons identified 2592 differentially expressed transcripts between 25 infants with symptomatic cCMV infection and healthy age-matched controls in the training set

---

**Table 1 Demographic data of healthy controls and infants with congenital CMV infection.**

| All | Healthy controls $n = 10$ | Symptomatic congenital CMV $n = 49$ | Asymptomatic congenital CMV $n = 31$ | *p* value |
|---|---|---|---|---|
| Age (days) | 11 (7–33) | 17 (11–22) | 17 (15–24) | 0.71 |
| Gestational age (weeks) | 38 (35–40) | 39 (38–40) | 39 (39–40) | 0.03 |
| Sex, males, *n* (%) | 7 (70%) | 30 (61%) | 16 (52%) | 0.52 |
| *Race, n* | | | | |
| White | 8 | 40 | 29 | 0.29 |
| Black or other* | 2 | 9* | 2 | |

Statistical analyses were performed using Kruskal–Wallis tests for non-parametric continuous variables and data reported as median, 25–75% interquartile ranges, and the Chi-square test for categorical data. Two-tail *p* values are provided.

**Table 2 Clinical information of infants with congenital cytomegalovirus infection.**

| | Symptomatic CMV $n = 49$ | Asymptomatic CMV $n = 31$ | P value |
|---|---|---|---|
| *Demographic information* | | | |
| Sex, males, n (%) | 30 (61%) | 16 (52%) | 0.49* |
| Gestational age (weeks) | 39 (38–40) | 39 (39–40) | 0.045 |
| Birth weight (grams) | 3035 (2457–3278) | 3195 (2951–3359) | 0.048 |
| Birth length (cm) | 48 (46–49) | 49 (47.4–50) | 0.024 |
| Head circumference (cm) | 33.4 (32–34.4) | 34 (33–34.6) | 0.021 |
| Age at sample collection (days) | 17 (11–22) | 17 (14–24) | 0.75* |
| *Examination findings, n (%)* | | | |
| Rash‡ | 16 (33%) | 0 | N/A |
| Splenomegaly | 12 (24%) | 0 | |
| Hepatomegaly | 11 (22%) | 0 | |
| SGA | 10 (20%) | 0 | |
| Microcephaly | 8 (16%) | 0 | |
| IUGR | 5 (10%) | 0 | |
| *Laboratory results (at diagnosis)* | | | |
| WBC (cells/mm$^3$) | 10,735 (8355–13,133) | 10,500 (9690–13,030) | 0.63 |
| Hemoglobin (g/dL) | 13.6 (11.3–17.2) | 13.9 (13.2–16.1) | 0.32 |
| Hematocrit (%) | 40.1 (33.4–48.4) | 39.3 (36.8–46.9) | 0.58 |
| Platelet count (/mm$^3$) | 257,000 (94,250–391,500) | 330,000 (251,500–417,500) | 0.049 |
| ALT (U/L) | 20 (14–31) | 17 (12.3–27.8) | 0.15 |
| Direct bilirubin (mg/dL) | 0.3 (0.2–0.53) | 0.3 (0.3–0.4) | 0.75 |
| *Neuroimaging findings, n (%)* | | | |
| Abnormal head ultrasound | 26 (58%) | 0 | N/A |
| Abnormal head CT | 3 (75%) | 0 | |
| Abnormal brain MRI | 8 (89%) | 0 | |
| *Audiologic findings, n (%)* | | | |
| Abnormal ABR at any time | 22 (45%) | 11 (35%) | 0.49 |
| Abnormal initial ABR | 9 (18%) | 0 (0%) | N/A |
| Abnormal follow-up ABR† | 13 (32%) | 11 (44%) | 0.43 |

Laboratory results available for 94% (46/49) infants with symptomatic congenital CMV infection and 94% (29/31) of asymptomatic CMV infection. Neuroimaging studies performed in 92% (45/49) of symptomatic and 90% (28/31) of asymptomatic CMV infection. Initial audiological evaluations performed in all infants with symptomatic or asymptomatic cCMV infection.
CMV cytomegalovirus, SGA small for gestational age, IUGR intrauterine growth restriction, WBC white blood cell, ALT alanine aminotransferase, CT computed tomography, MRI magnetic resonance imaging, ABR auditory brainstem response. Continues variables reported as medians and 25–75% interquartile range. Statistical comparisons computed by two-tailed Mann–Whitney or Fisher's T test (*).
‡Eight patients with petechiae, four patients with purpura, two patients with a "blueberry muffin" rash, and three patients with a combination of above.
†Follow-up evaluation of at least 1 year. Follow-up ABR available for 41 infants with symptomatic CMV infection and 25 infants with asymptomatic CMV infection.

(symptomatic cCMV signature; Fig. 1a). Of those, 95% of transcripts were overexpressed and 5% were underexpressed. The top 10 overexpressed transcripts were related to interferon (IFI44/L, IFI44, IFIT1, OAS3), T cells (LAG3–lymphocyte activation gene–), or granzymes (GZMH). In contrast, the underexpressed transcripts were related to hemoglobin (HBE1), myeloid cells (TREML3P-triggering receptor expressed on myeloid cells-) or other genes involved in cell trafficking and signaling, including: a brain expressed protein (BEX1), membrane-associated ring finger protein (MARCHF2), or calcium-binding protein (CABP5) (Supplementary Table 4). This 2592 gene biosignature was validated by principal component analyses (PCA) in an independent cohort of 24 symptomatic CMV-infected infants and the same healthy controls used in the derivation cohort (validation cohort; Fig. 1b; Supplementary Fig. 2).

A similar approached was followed with the asymptomatic cohort, and the asymptomatic cCMV biosignature was derived in a third group of infants (training set, $n = 16$), which was validated in a fourth independent group of asymptomatic cCMV infants (test set, $n = 15$). Statistical group comparisons identified 3324 differentially regulated transcripts between 16 asymptomatic cCMV infants and healthy controls in the training set (asymptomatic cCMV signature; Fig. 1c). This signature was also validated by PCA in an independent cohort of infants with asymptomatic cCMV and 10 healthy controls (validation cohort; Fig. 1d). The top 10 over- and underexpressed transcripts of the asymptomatic cCMV signature shared common genes with the symptomatic

profiles including overexpression of interferon related genes (IFI44/L, IFI44, IFIT), LAG3 and GZMH, or the underexpressed HBE1 gene (Supplementary Table 4).

To ensure the reproducibility of the biosignatures and assess potential selection bias in infants included in the derivation cohorts, the training (discovery) and test (validation) sets for infants with symptomatic and asymptomatic cCMV were derived separately. Correlation analysis of this data showed a high degree of correlation between the test and training sets for symptomatic (Spearman $r = 0.88$; $p < 0.0001$) and asymptomatic cCMV infection (Spearman $r = 0.94$; $p < 0.0001$). (Supplementary Fig. 3).

The overlap between the symptomatic and asymptomatic cCMV signatures was further evaluated, with 58% of transcripts (2160 transcripts) shared among both signatures (Supplementary Fig. 4). Among the top 10 over- and underexpressed transcripts unique for the symptomatic cCMV signature, OTOF, KIR2DL1, and CCZ1 were overexpressed, and TREML1 underexpressed. Similarly, of the unique transcripts for the asymptomatic cCMV signature, the top 10 over- and underexpressed transcripts included overexpression of CLECL1 and IDO1.

To determine the ability of the symptomatic cCMV biosignature to discriminate between infants with symptomatic and asymptomatic infection, the symptomatic biosignature was applied to the entire cohort of infants with congenital CMV infection ($n = 80$) and healthy controls. Unsupervised cluster analyses yielded a mixed distribution that did not reliably separate asymptomatic from symptomatic CMV infection (Fig. 2a). A similar result was

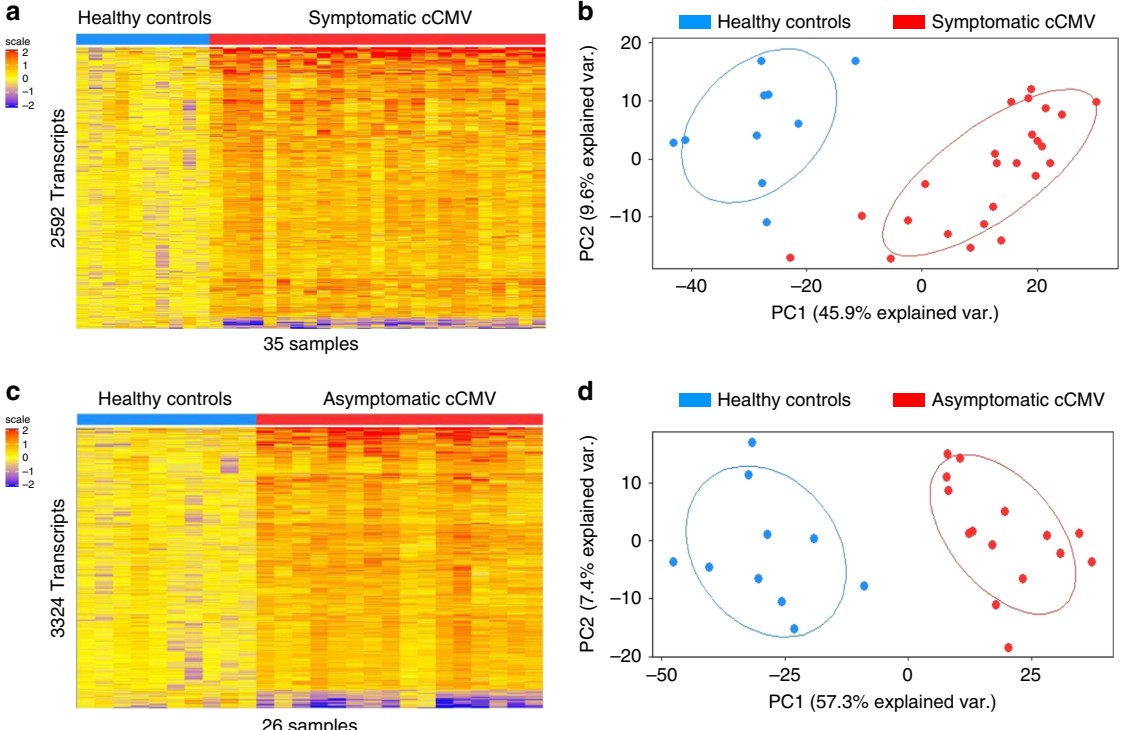

**Fig. 1 Biosignatures for symptomatic and asymptomatic cCMV infection. a** Statistical group comparisons using linear models (LIMA) adjusted for age followed by Benjamini–Hochberg FDR < 0.01 and ≥1.5 fold change between infants with symptomatic cCMV infection ($n = 25$) and healthy controls ($n = 10$) identified 2592 differentially expressed transcripts in the training set (symptomatic cCMV biosignature). **b** The transcriptional signature was validated by principal component analyses (PCA) in an independent test set of 24 infants with symptomatic cCMV and the same 10 healthy controls. Healthy controls are represented in blue and symptomatic cCMV patients in red. **c** A similar strategy was applied to derive and validate the asymptomatic cCMV biosignature using 16 cCMV infants in the training set and 10 healthy controls, which yielded 3324 differentially expressed genes. **d** The asymptomatic cCMV biosignature was validated by PCA in an independent test set of 15 infants with asymptomatic cCMV infection and the same healthy controls. Transcripts in **a** and **c** are organized in a heatmap format where each row represents a transcript and each column a patient sample. Red color indicates overexpression and blue color underexpression of a transcript compared with the median expression of healthy controls (yellow).

observed when the asymptomatic cCMV biosignature was applied to the entire cohort (Fig. 2b). Thus, although in both analyses all 10 healthy controls clustered separately from cCMV-infected infants, neither the asymptomatic nor symptomatic cCMV biosignatures were able to reliably distinguish cCMV-infected infants based on their clinical classification.

**Modular analysis of infants with cCMV infection.** To characterize the biological significance of the symptomatic and asymptomatic biosignatures, an analytical framework of 62 transcriptional modules was applied. Each module (M) consists of coordinately expressed genes that share a similar biological function[16,17]. Of the 62 modules analyzed, 16 related to innate and adaptive immune responses are represented in Fig. 3. Modular maps were derived independently for the discovery (training sets) and validation cohorts (test sets) of infants with symptomatic and asymptomatic cCMV infection in relation to the healthy control group. For reproducibility, results were compared between the training and test sets in each clinical condition (symptomatic and asymptomatic cCMV infection) and between infants with symptomatic and asymptomatic cCMV infection.

Modular maps for infants with both symptomatic and asymptomatic cCMV infection showed overexpression of modules related to interferon, T cells, B cells, plasma cells, and cytotoxic/NK cells (Fig. 3, Supplementary Table 6), with no significant differences between the two groups ($p > 0.05$; Fisher's exact test and Bonferroni multiple test corrections). In addition, modules related to monocytes and inflammation, were both underexpressed in symptomatic and

asymptomatic cCMV infection, also with no significant differences between groups. These findings were further validated between the training and the test sets of the symptomatic (Supplementary Fig. 5) and asymptomatic cohorts, respectively, (Supplementary Fig. 6). As an additional validation step, we compared the modular maps of the discovery cohorts of infants with symptomatic and asymptomatic cCMV infection and again demonstrated significant correlations between both cohorts (Spearman $r = 0.93$, $p < 0.0001$; Supplementary Fig. 7). Thus, using two different analytical strategies, we identified and validated the symptomatic and asymptomatic cCMV transcriptional signatures at the gene level (Fig. 1), and at the modular level (Fig. 3) in independent sets of patients, and showed a high degree of similarity between infants with cCMV infection irrespective of their clinical classification.

**Molecular distance to health scores in cCMV infection.** To investigate whether global transcriptional differences allowed discrimination between infants with symptomatic and asymptomatic cCMV infection, we calculated the molecular distance to health (MDTH) genomic score. This metric summarizes into a numeric value the global transcriptional perturbation of each individual patient sample compared with age-matched healthy controls[11,18–20]. To calculate the MDTH scores, 3756 transcripts identified in either the symptomatic or asymptomatic cCMV biosignatures were utilized (Supplementary Fig. 3). Overall, MDTH scores were significantly higher in infants with cCMV infection compared with healthy controls irrespective of their clinical classification. However, no significant differences in

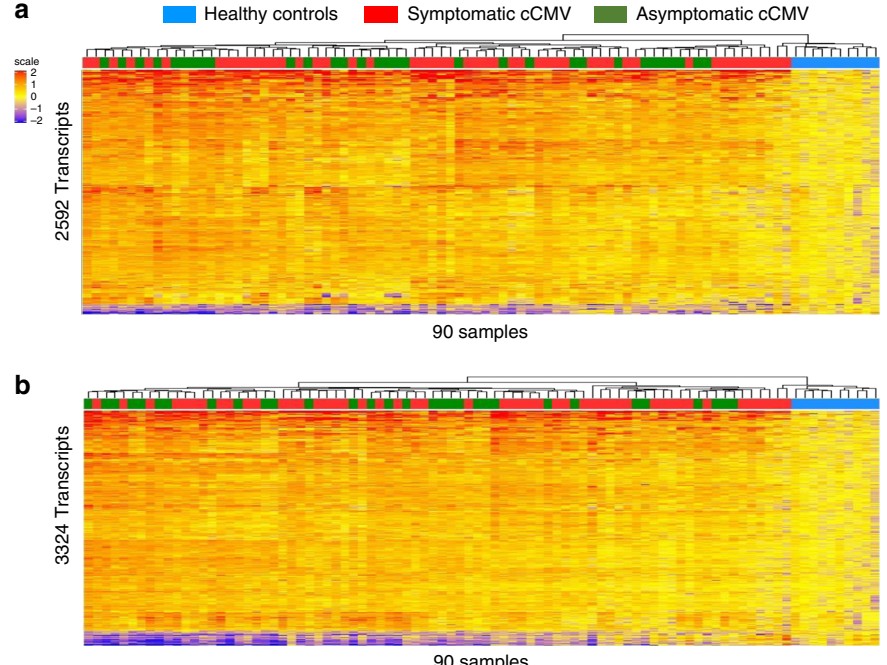

**Fig. 2 Biosignatures fail to distinguish symptomatic or asymptomatic cCMV infection. a** Using unsupervised hierarchical clustering, the symptomatic CMV biosignature (2592 transcripts) applied to the entire cohort of infants with CMV ($n = 80$) and 10 healthy controls, yielded a mixed distribution that did not reliably distinguish symptomatic from asymptomatic cCMV infection. **b** Following the same strategy, the asymptomatic CMV biosignature (3324 transcripts) was applied to the entire cohort of infants utilizing unsupervised hierarchical clustering and also yielded a mixed distribution that did not reliably distinguish asymptomatic from symptomatic cCMV-infected infants. Transcripts are organized in a heatmap format where each row represents a transcript and each column represents a patient sample. Red color indicates overexpression and blue color underexpression of a transcript compared with the median expression of healthy controls (yellow). Clusters for healthy controls are depicted in blue, for infants with symptomatic cCMV infection in red and asymptomatic cCMV infection in green.

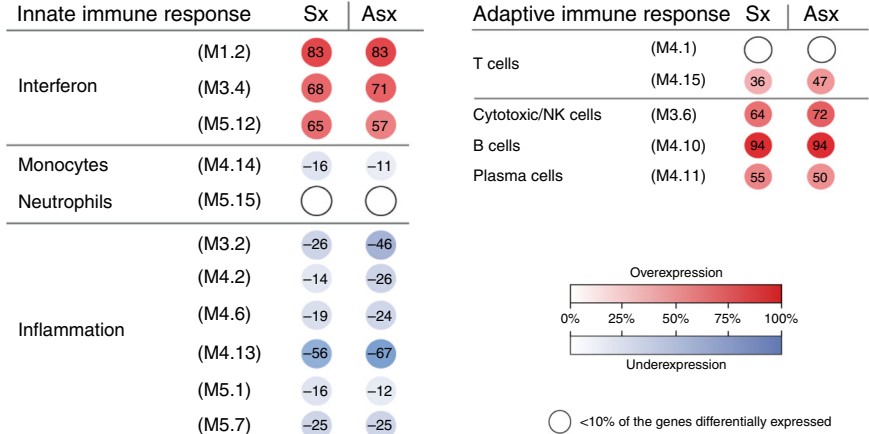

**Fig. 3 Modular expression in symptomatic and asymptomatic cCMV infection.** Modules (M) are groups of genes that are co-expressed and shared a similar biological function. Sixteen selected modules pertaining to innate and adaptive immune responses are represented above and included three related to interferon, one to monocytes, one to neutrophils, six to inflammation, two T cell related, one related to cytotoxic/NK cells, one plasma cell, and one B-cell related. The intensity of the modules (dots) indicate the proportion of overexpressed (in red) or underexpressed (in blue) transcripts within each module. Numeric values indicate the percentage of transcripts expressed in each specific module compared with healthy controls. Blank dots indicate that <10% of the genes in the module were differentially expressed compared with controls. The panel on the left includes innate immunity-related modules, and the panel in the right is comprised of adaptive immunity modules. Sx symptomatic, Asx asymptomatic.

MDTH scores were observed between symptomatic or asymptomatic cCMV-infected infants (Fig. 4).

**Longitudinal transcriptional analysis of cCMV infection.** Twenty-three infants included in the symptomatic cCMV cohort and 15 in the asymptomatic cCMV cohort had at least one sample obtained at follow-up visits at one, two, or three years of age. To be able to map changes in initial gene expression over time, the symptomatic (2592 transcripts; Fig. 5a) and asymptomatic (3324 transcripts; Fig. 5b) cCMV biosignatures were

applied to the longitudinal samples within each cohort that showed that the initial changes in overexpression of transcripts persisted up to 3 years of age.

Modular analyses were also applied to the longitudinal samples and revealed that while initial inflammation transcripts were underexpressed in infants with symptomatic and asymptomatic cCMV infection, expression levels normalized in year one and were significantly overexpressed in infants with asymptomatic vs symptomatic cCMV infection ($p < 0.009$ Fisher's exact test and Bonferroni multiple test corrections) at 2 years of age, and remained overexpressed in both clinical groups thereafter (Fig. 6). Modular neutrophil expression changed during the first year of age from neutral/mildly underexpressed to greatly underexpressed, and remained as such thereafter, whereas monocyte-related genes also became underexpressed but later returned to their initial baseline. Overexpression of interferon and T-cell genes were observed initially in both asymptomatic and symptomatic infants, though both declined with time. B-cell and plasma cell modular overexpression were identified during the first year of age and plateaued at subsequent follow-up visits. Thus, except for inflammation related genes, analyses of

longitudinal samples revealed similar trends for modular gene expression over time irrespective of the initial clinical disease classification.

**Transcriptional profiles of late-onset SNHL in cCMV.** Given the potential clinical impact of a biomarker associated with SNHL in cCMV infection, we sought to identify classifier genes at the time of diagnosis that were associated with the development of late-onset SNHL. Overall, 24 cCMV infants passed the newborn hearing screening (11 asymptomatic and 13 symptomatic) and went onto developing SNHL at any time point during the 3-year follow-up period. The initial transcriptional profiles (samples obtained at enrollment) from these 24 infants were compared with those from 28 cCMV-infected children (12 asymptomatic and 16 symptomatic) that did not develop SNHL during the follow-up period and had at least 900 days of follow-up. Of the 11 infants with asymptomatic cCMV that developed SNHL, 7 were bilateral, 4 were unilateral, and were of mild ($n = 7$) or moderate ($n = 3$) degree. The severity in one patient was not reported. Of the 13 infants with symptomatic cCMV, SNHL was bilateral in 8 children and unilateral in 5, and were of mild (11) or severe (2) degree. The demographic characteristics of these infants are shown in Supplementary Table 7.

Random Forest-Recursive Feature Elimination (RF-RFE) analyses applied to samples obtained at the time of cCMV diagnosis in these 52 cCMV infants (24 who developed late-onset SNHL and 28 who did not), identified 16 classifier genes (Supplementary Table 8) that were associated with late-onset SNHL with 92% accuracy, and an area under the curve of 0.97 (Supplementary Fig. 8). Median expression values for those genes in both groups of infants with or without late-onset SNHL are displayed in Fig. 7a. Of the 16 genes, *CD40*, *RAB9B*, and *MATR3* have been associated with innate immune related processes, whereas *ARHGEF9* and *MPDU1* have been associated previously with intellectual disability[21,22]. Validation of this 16-gene signature by PCA showed separation at baseline in infants with cCMV according to the development of late-onset SNHL (Fig. 7b).

## Discussion

In this study, infants with symptomatic and asymptomatic cCMV infection exhibited distinct blood biosignatures compared with healthy controls. However, these biosignatures were unexpectedly similar and did not discriminate between infants with clinically apparent disease and those who were clinically normal. Importantly, and although validation in additional cohorts is needed, we

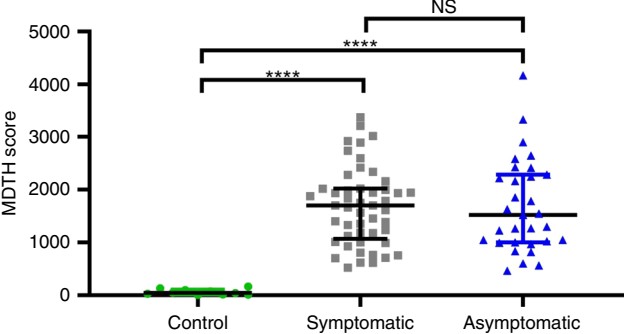

**Fig. 4 MDTH scores are similar in symptomatic or asymptomatic cCMV infection.** Transcripts that were differentially expressed in either symptomatic or asymptomatic cCMV infection (3756 transcripts) were used to calculate the molecular distance to health (MDTH) scores for healthy controls ($n = 10$), symptomatic ($n = 49$), and asymptomatic ($n = 31$) cCMV infection. Data are represented as individual MDTH scores per patient (green circles: healthy controls; gray squares: symptomatic cCMV infection; blue triangles: asymptomatic cCMV infection) along with medians and 25–75% interquartile ranges. Groups comparisons were performed using Kruskal–Wallis with Dunn's test for multiple comparisons; **** two tailed $p < 0.0001$; NS non-significant.

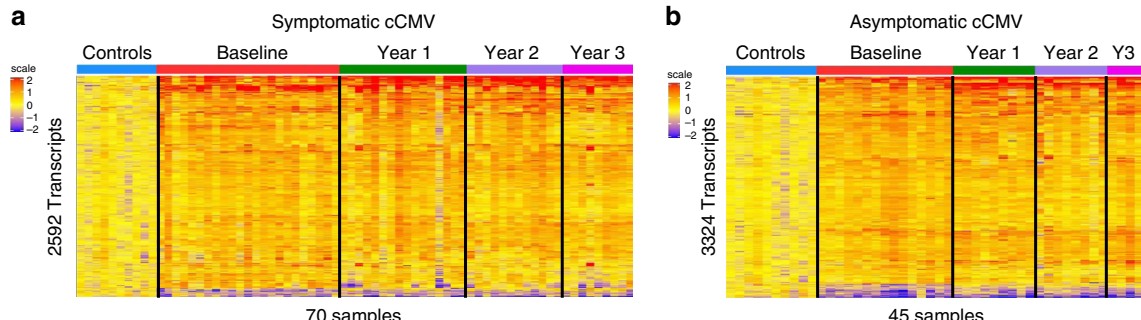

**Fig. 5 Longitudinal transcriptional analysis of infants with cCMV infection. a** The symptomatic biosignature (2592 transcripts) was applied to symptomatic cCMV-infected infants with samples collected at initial testing ($n = 23$), 1 year ($n = 16$), 2 years ($n = 12$) and 3 years ($n = 9$) of follow-up. **b** The asymptomatic cCMV biosignature (3324 transcripts) was also applied to asymptomatic cCMV infants with samples collected at initial testing ($n = 15$), 1 year ($n = 9$), 2 years ($n = 8$), and 3 years ($n = 4$) of follow-up. Transcripts were organized in a heatmap format where each row represents a transcript and each column represents a patient sample. Red color indicates overexpression and blue color underexpression of a transcript compared with the median expression of healthy controls (yellow).

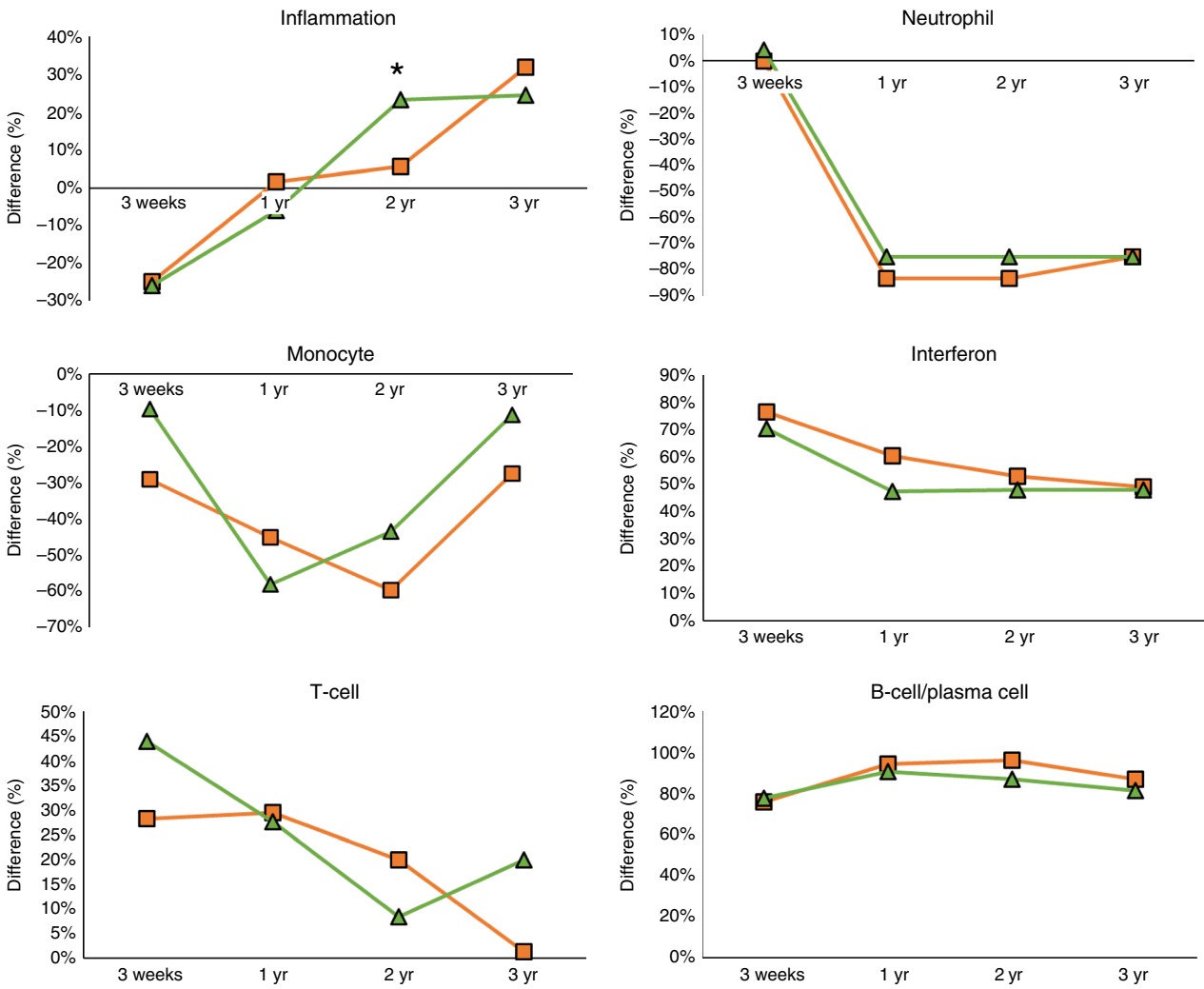

**Fig. 6 Longitudinal modular analyses of infants with cCMV infection.** Mean percent difference* in modular expression of innate (inflammation, neutrophil, monocytes, interferon) and adaptive immunity (T cell and B cell/plasma cell) modules are plotted and compared over time (baseline, 1 year, 2 years, and 3 years of age) for infants with symptomatic (orange square) and asymptomatic (green diamond) congenital CMV infection. *The mean percent difference was calculated by the percentage of genes upregulated (positive percent) minus the percentage of downregulated genes (negative percent) within each module. When the biological functions were represented by more than one module (i.e., interferon, inflammation, T cells, see Fig. 3), modular mean was derived and the percentage difference calculated as explained above.

identified a 16-gene signature at cCMV diagnosis that was associated with the development of late-onset SNHL, suggesting its potential value as a biomarker.

CMV has the ability to infect multiple organs in utero, resulting in a wide spectrum of disease manifestations at birth that range from clinically inapparent infection to central nervous system disease with severe global neurodevelopmental delay and SNHL. Despite the wide range of clinical manifestations, the determinants that drive the observed symptomatology (or lack thereof) are largely unknown. Likewise, there are no factors, including blood CMV loads, that have shown to reliably predict the development of late-onset SNHL in both symptomatic and asymptomatic infants[9,23–27]. Furthermore, although infants with symptomatic disease demonstrate higher rates of SNHL than those that are asymptomatic, the subsets within both symptomatic and asymptomatic infants who are at highest risk of developing SNHL are not well defined[28]. Our first goal was to determine whether transcriptional profiles in infants with cCMV infection could distinguish those with symptomatic and asymptomatic disease. Despite applying a number of analytical strategies, we were unable to differentiate infants with symptomatic or asymptomatic cCMV infection at the gene and modular level or by applying the genomic molecular distance to health score. This was an unexpected observation and suggests that the host transcriptional immune response is similar in infants with cCMV infection irrespective of their clinical presentation and supports the premise that congenital CMV infection is a spectrum of clinical disease presentation as opposed to discrete entities. The tremendous overlap between conditions emphasize the complex nature of this chronic infection. Although the asymptomatic infant with cCMV infection may not have overt clinical findings of disease, the patient's immune response suggests otherwise and perhaps the definition of the asymptomatic infant may need to be revised.

CMV infection is mostly controlled by CD4+ and CD8+ T-cell responses, and impaired T-cell immunity in infants with cCMV infection has been reported[29–32]. Expansion of CMV-specific CD4+ and CD8+ T cells have been described in infants with cCMV infection; however, their ability to generate robust cytokine responses were impaired relative to the adult CMV immune response[33,34]. In agreement with those findings, we found that T-cell and B-cell-related transcripts were strongly

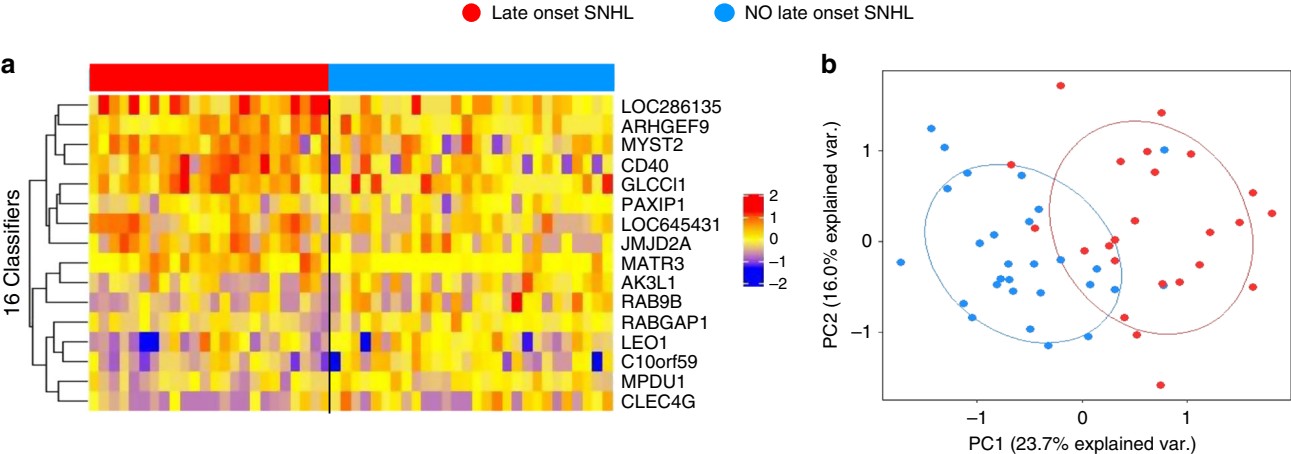

**Fig. 7 Classifier genes for late-onset SNHL in infants with cCMV infection.** Twenty-four infants with cCMV infection (11 asymptomatic and 13 symptomatic) developed SNHL during follow-up compared with 28 cCMV infants (12 asymptomatic and 16 symptomatic) that did not develop SNHL and had at least 900 days of follow-up. **a** Random Forest-Recursive Feature Elimination (RF-RFE) algorithm, identified a 16-gene signature that classified cCMV-infected infants who developed SNHL with 92% accuracy with samples obtained at baseline. The 16-gene list signature is displayed in a heat map format where each column represents the profile of an infant and each row one of the 16 transcripts. Red indicates overexpression and blue transcript underexpression. **b** The 16 classifier gene set was validated by principal component analyses (PCA) that showed two clusters of cCMV infants according to the development of late-onset SNHL.

overexpressed, in both symptomatic and asymptomatic infants with cCMV infection. Apart from T-cell responses, interferon responses are essential for the modulation and control of viral infections, and robust IFN-γ responses have been reported in cCMV-infected infants in utero[35]. In our cohort, we observed a comparable and strong overexpression of interferon-stimulated genes in both symptomatic and asymptomatic cCMV infants. Notably, increased expression of *IFI44L*, *IFTI1,* and *IFI44* were among the top overexpressed genes in both groups of infants irrespective of their clinical classification. Contrary to the immune profiles described in children with a number of acute infections, which showed overexpression of innate immunity genes and lack of activation or suppression of adaptive immunity[11–13], infants with cCMV infection showed enhanced expression of adaptive immunity genes with relative suppression of inflammation and monocyte-related genes. As cCMV infection occurs in utero, this is not unexpected as adaptive immune responses have been detected in utero of congenitally infected infants[35]. Interestingly, when we applied the biosignatures of symptomatic and asymptomatic cCMV infection to longitudinal samples, the abnormal immune profiles identified during the first 3 weeks of life persisted for years after initial testing, likely reflecting a persistent viral infection that leads to a chronic stimulation of the immune system.

Despite numerous studies addressing the role of blood CMV PCR as potential indicator of SNHL[9,27,36], no biomarkers have been able to reliably predict late-onset SNHL in either symptomatic or asymptomatic cCMV-infected infants. In a recent study, Rovito et al.[15] analyzed blood transcriptional profiles that were derived from blood dried blood spots in 12 infants with cCMV infection and 6 healthy controls. Although, no significant differences in gene expression were identified between groups, likely related to the small sample size, *LAG3*, *IFIT1*, *OAS3*, or *GZMH*, were overexpressed in their study, and also among our patients with cCMV infection, further validating our findings[15]. By applying a Random Forest classification algorithm to infants with cCMV infection and normal hearing in the newborn period, we identified a group of 16 genes that were associated with the development of late-onset SNHL with 92% accuracy. Among these 16 genes, *CD40* has been found to be increased in patients with SNHL[37], and *ARHGEF9* and *MPDU1* in patients with

intellectual disability[21,22]. No overlap was noted between the 16-gene classifier set and those identified by Rovito et al.[15], which may be explained by differences in characteristics among cohorts or methodologic differences. On the one hand, dry blood spots offer the advantage of leveraging samples routinely obtained in a standardized manner as part of the newborn screening however, mRNA degradation in samples collected prospectively for transcriptome analyses is less likely to occur, thus offering higher sensitivity. Although these data are encouraging, validation in larger cohorts of patients with adequate follow-up is necessary, as identification of an early signature could facilitate targeted therapies for infants with cCMV infection, while also identifying those who would benefit from additional therapeutic interventions (i.e., hearing aids, speech therapy).

Our study has limitations. First, patient samples were collected in the first 21 days of age. Although there were no differences between healthy controls and infants with cCMV infection in terms of basic demographic characteristics, it is difficult to exclude possible changes in gene expression that could be related to birth and transition to extra-uterine life. Nevertheless, the biosignature of cCMV was suggestive of a chronic infection that persisted over time. The same healthy controls were used throughout all analyses and thus, an independent healthy control cohort was not included with the validation sets. Similarly, for the longitudinal analyses, the same healthy young infant controls were used for the three follow-up time points. Although analyses of those time points did not include age-matched controls, our approach allowed for the initial transcriptional profiling (early infancy) to serve as a reference value over time, and suggests that the biosignature of cCMV is one of a chronic infection. Although not ideal, the challenge of enrolling healthy controls (particularly young infants) has led to similar limitations in prior studies with consistent results[12,13]. Quantitative blood CMV PCR was not performed routinely, and thus we are unable to compute correlations between CMV loads and transcriptional data. Similarly, because of limitations in blood samples volume, we did not validate the data at the protein level or performed functional assays to correlate transcriptional profiles and functional immune responses, and those studies should be conducted in the future. The limited number of samples at follow-up reduced the ability to perform more robust comparisons, most notably with limitations

in the number of asymptomatic infants with cCMV who did and did not developed SNHL. In addition, the cohort of asymptomatic cCMV infants enrolled developed SNHL at higher rates than previously reported[4], and thus it may reflect a biased population with greater disease severity, limiting generalizability. Despite these limitations, and the need for validation in independent cohorts, we were able to shed light into the pathogenesis of cCMV in infants and identify a set of biomarkers associated with late-onset SNHL.

In summary, despite differences in clinical, laboratory, and neuroimaging findings, asymptomatic, and symptomatic cCMV-infected infants demonstrated similar host transcriptional immune profiles. Thus, cCMV infection likely represents a broad continuum rather than discrete entities of symptomatic and asymptomatic disease. In addition, we identified a group of genes that were associated with subsequent development of late-onset SNHL in infants with cCMV infection. Confirmatory analyses are needed to validate the value of this (or similar) signature as a potential biomarker of late-onset hearing loss.

## Methods

**Study design**. This was a prospective cohort study of infants with cCMV infection and healthy controls who were enrolled at Parkland Memorial Hospital and Children's Medical Center, Dallas, TX and Nationwide Children's Hospital, Columbus OH, from October 2006 to April 2013. cCMV infection was defined as a positive culture or PCR test of urine (88%) and/or a positive PCR from saliva (12%) samples obtained in the first 21 days of age. Positive saliva samples were all confirmed by CMV urine culture or PCR. The real-time PCR assay included probes and primers targeting a highly conserved region of the envelope glycoprotein B (AD-1 region) and a highly conserved immediate early 2 exon 5 region as described elsewhere[38,39].

Children with cCMV infection were evaluated sequentially during the first three years of age for the development of SHNL. A blood sample for transcriptional profile analyses and for complete blood cell count (CBC) with differential, along with clinical information, were obtained at enrollment, and at 1, 2, and 3 years of age. In parallel, we enrolled a cohort of healthy age-, gender-, and race-matched controls at well-child visits or prior to undergoing minor elective surgical procedures. Healthy controls were excluded from the study if they had an acute illness, exposure to antibiotics or steroids within 2-weeks of enrollment, or any underlying comorbidity. The study was approved by the IRBs of the University of Texas Southwestern Medical Center, Dallas, TX and Nationwide Children's Hospital in Columbus, OH, USA and written informed consent obtained from parents/legal guardians before study enrollment.

Infants with symptomatic cCMV infection were identified by targeted CMV screening if they had clinical or laboratory signs consistent with CMV infection, referred on the newborn hearing screen, or had additional risk factors including: small for gestational age, defined as birth weight <10%, intrauterine growth restriction defined as a ponderal index <10%, born to mothers infected with HIV, or infants of <34 weeks' postmenstrual age due to the inability of performing hearing screening at an earlier gestational age)[40]. Infants with asymptomatic cCMV infection were identified through the CMV & Hearing Multicenter Screening (CHIMES) study, and blood for transcriptome analyses obtained under a separate IRB-approved study protocol (see above)[38]. Symptomatic and asymptomatic infants with cCMV infection underwent a complete evaluation that included physical examination, laboratory, audiologic, ophthalmologic, and radiologic studies[41]. Specifically, results of CBC and platelets, serum alanine aminotransferase and bilirubin (total/direct) concentrations, eye examination, cranial ultrasonography (or other neuroimaging studies), auditory brainstem evoked responses (BSER), and results from neurodevelopmentally appropriate behavioral auditory evaluations were recorded.

**Study definitions**. Symptomatic cCMV infection was defined by any abnormality identified on (a) physical examination (hepatomegaly, splenomegaly, skin rashes (petechial, blueberry muffin, or purpura), and microcephaly defined as a head circumference <10% for gestational age); (b) laboratory testing including anemia (hematocrit < 35%), thrombocytopenia (platelet count of <150,000 mm³), direct hyperbilirubinemia (>2 mg/dL) or increase transaminases (ALT ≥ 40 U/mL for term newborns born at ≥37 weeks' gestation, and ALT > 45 U/mL if <37 week's gestation); (c) neuroimaging (lenticulostriate vasculopathy, periventricular calcifications, cortical dysplasia); (d) ophthalmologic examination, or (e) hearing evaluation at birth. Infants who did not have any of these findings were classified as asymptomatic[41]. Late-onset SNHL was defined as the presence of a normal newborn hearing evaluation followed by an abnormal hearing evaluation (BSER) at any of the follow-up evaluations. Auditory BSER and behavioral audiologic thresholds of 0–20 dB were considered normal hearing, whereas thresholds of 21–30, 31–60,

and 61–90 dB constituted mild, moderate, and severe hearing loss, respectively. Findings of mild hearing loss or greater were considered abnormal.

**CMV real-time PCR**. Detection of CMV DNA was performed using the ABI 7500 real-time PCR System (Applied Biosystems Inc, Foster City, CA) and ABsolute QPCR Low ROX Mix (ABgene USA, Rockford, IL), and concentrations of primers and probes in the reaction mixture were 900 and 250 nM, respectively as described. The amplification conditions have been described elsewhere[38,39]. In brief, samples that underwent PCR testing were run in duplicates using 25 μL of the reaction mixture (20 μL of master mix and 5 μL of sample). Standard curves were generated in each plate using plasmid standards incorporating the target sequences in 10-fold dilutions, ranging from 100,000 and 10 genomic equivalents per reaction. The real-time PCR assay used included two sets of probes and primers. The first primer assay targeted a highly conserved regions of the envelope glycoprotein B (AD-1 region)[39]. The forward primer was 5′-AGGTCTTCAAGGAACTCAGCAAGA, and the reverse primer was 5′-CGGCAATCGGTTTGTTGTAAA. The internal probe 5′-ACCCCGTCAGCCATTCTCTCGGC was labeled at the 5′-end with fluorescent dye 6-carboxyfluorescein (i.e., FAM), as the reporter dye, and the 3′-end was labeled with quencher dye 6-carboxytetramethylrhodamine (i.e., TAMRA). The two-primer assay targeted a highly conserved immediate early 2 exon 5 region (forward primer, GAGCCCGACTTTACCATCCA; reverse primer, CAGCCGGCGGTATCGA; and probe VIC-ACCGCAACAAGATT-MGBNFQ[38]. The detection limit of the PCR assay as determined by sensitivity titration was 250 to 50 genomic equivalents per mL depending on the single or two-primer assay used respectively[38,39].

**Sample collection and processing**. Blood samples (200 μL⁻¹ mL) were collected in Tempus tubes adapted for small blood volumes (Applied Biosystems, CA, USA) at the time of diagnosis and follow-up visits, and stored at −20 °C within 12 h of collection until further processing in batches[12]. Whole-blood RNA was processed and hybridized into Illumina Human HT12 V4 beadchips (47,323 probes) and scanned on the Illumina Beadstation 500[11].

**Microarray data and statistical analysis**. For purpose of analyses, a stepwise approach was followed. The biosignatures of infants with symptomatic and asymptomatic cCMV infection were first derived and then validated separately. As second step we assessed the longitudinal evolution of the biosignatures over the 3 years of life. Finally, we identified the transcriptional profiles associated with development of late-onset SNHL. JMP genomics (version 8.1) and R software (version 3.6.0) packages were used for analyses purposes[13]. In brief, transcripts were first selected if they were present in >10% of all samples and had a minimum of twofold expression change compared with the median expression across all samples (quality control-QC gene list). The following strategy was then applied[11–13]: (a) supervised analysis (comparative analyses between predefined groups) was performed using linear models (LIMMA package 3.42 in R) adjusted for age, followed by Benjamini–Hochberg multiple test correction (false discovery rate 1%) and a >1.5 fold change filter in expression level relative to the healthy control group; (b) unsupervised clustering by PCA was used for validation purposes; (c) functional gene analyses were performed using modular analysis. In brief, modular analysis is a systems scale strategy that aims to reduce the abundance of transcriptional data into functional pathways. This approach uses clusters of co-expressed genes (or modules) to generate disease-specific transcriptional finger-prints, providing a stable framework for the visualization and functional inter-pretation of gene expression data[10,16,17]. Modular maps were generated using a stepwise approach and visualized in a grid format, where the first round of modules (M1) is represented by the sub-network with the most genes that are co-clustered in all input data sets. In the next rounds of selection, the level of stringency to identify the core networks is relaxed, so modules are formed by genes that co-cluster in all but one of the data sets (M2), two of all the datasets (M3) and so on. For visualization purposes, the significant abundance of transcripts relative to a baseline (or healthy controls) are represented by a colored dot. When the pro-portion of overexpressed transcripts in a given module is increased, the module is represented by a red dot, whereas an increased proportion of underexpressed transcripts is represented by a blue dot, with the intensity of the color, indicating the proportion of transcripts expressed in any given module[10,15–17,41]; (d) MDTH, a tool that converts the global transcriptional perturbation of each individual patient sample into an objective score in relation to the healthy control baseline[11,18–20], was calculated and compared between symptomatic and asymp-tomatic infants with cCMV infection and healthy controls.

To evaluate whether gene expression profiles identified in the neonatal period evolved over time, the biosignatures derived for infants with symptomatic and asymptomatic cCMV infection at baseline were applied to the longitudinal patient samples obtained on year 1, 2, and 3. Last, a RF-RFE algorithm was applied to gene expression data and clinical variables, including presence of microcephaly and abnormal neuroimaging findings at CMV diagnosis, to identify classifiers associated with the development of SNHL[42]. The analysis was performed in R environment with "caret" package. Two clinical variables (microcephaly and neuroimaging) were included for feature selection which started with 3/4 of samples that were selected randomly as the training set and all the genes were used

for feature selection with 10-fold internal cross validation. Next, the prediction model was applied to the rest 1/4 of samples for external validation to assess prediction accuracy. RF-RFE method was repeated 10 times and the best random forest prediction model with highest accuracy was selected.

Statistical analyses for clinical variables were performed using Graph Pad Prism V6 (San Diego, CA). In brief, non-parametric tests (either Mann–Whitney $U$ or Kruskal–Wallis) were used to evaluate differences in continuous variables between groups, whereas differences in proportions were assessed using Fisher's exact and chi-square test as appropriate. All tests were two-tailed with $p$ value < 0.05 considered statistically significant.

**Reporting summary**. Further information on research design is available in the Nature Research Reporting Summary linked to this article.

## Data availability

Microarray data that support the findings of this study have been deposited in the NCBI Gene Expression Omnibus with the primary accession number GSE108211 [https://www.ncbi.nlm.nih.gov/geo/query/acc.cgi?acc=GSE108211]. All figures presented in this manuscript are generated from the raw data as provided in the Source Data file.

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

## Acknowledgements

We would like to Cynthia Smitherman and Phuong Nguyen at the microarray core at the Baylor Institute for Immunology Research, Dallas, TX for their help with RNA processing and hybridization, and especially to our patients and their families for agreeing to participate in the study. This work was supported by intramural grants including the Grant consortium #20054914 at Nationwide Children's Hospital to C.P.O., A.M., and P.J.S. A.R. received grant support from "A. Griffini–J. Miglierina" Fundation, Varese-Italy.

## Author contributions

A.M., P.J.S., and O.R. designed and oversaw the study; Z.X., D.B., D.C., M.S., and C.P.O. conducted data analyses and interpretation; F.K., A.R., L.L., K.E.O., and A.G.S. enrolled patients, collected and analyzed patient information. All authors contributed to the final manuscript.

## Competing interests

The authors declare no competing interests.
