## [Peer Review File · Nature Communications]

Reviewers' comments:

Reviewer #1 (Remarks to the Author):

This manuscript describes the blood genome expression profiles in neonatal whole blood in children with congenital CMV infection, and compares the profiles of asymptomatic cCMV children as well as symptomatic cCMV children with a healthy control group. Furthermore, both cCMV groups are compared to each other. This manuscript is well written, the way the experiments are performed (with a training and a test panel) is thorough and leads to robust conclusions in this specific cohort. The conclusions, that the expression profiles in asymptomatic and symptomatic children are comparable and non-distinguishable is very interesting. The information on functional genes gives insight into pathogenetic mechanisms and the authors have identified a set of classifier genes that could potentially predict which infants within the asymptomatic group have increased risk of developing late-onset hearing loss. If this can be confirmed in another cohort, this will indeed impact the clinical policy in these children.

I have to mention that I am not an expert on statistical analyses of gene expression data. I do think the authors are clear and transparent in explaining their statistical methods.

There are three points that need further attention:

1. There is a previous paper on expression profiles in neonatal whole blood on filter paper in children with cCMV and a healthy control group. This should be included in this manuscript.
2. The follow-up samples are very interesting, also in view of the longer term pathogenesis. However, it is not clear to me, whether the expression profiles at the different time points were compared to the neonatal healthy control group, or to follow-up samples of the healthy control group at similar time points. This may impact the results and conclusion.
3. The authors should acknowledge in the discussion that their cohort reflects a more severely affected cohort than you would expect to find through universal screening. This is important because it may influence the applicability of e.g. the classifier-gene set.

Comments:

Introduction:

- Page 3, lines 60-67: The percentage of children that develops SNHL in the symptomatic cCMV is not 50 %, but somewhat lower. The systematic review of Goderis et al. (2014) states 1 in 3 symptomatic children will experience SNHL. The group of Fowler showed 40 % SNHL in symptomatic children and in the paper of Foulon et al. (2019) 44 % had SNHL (in a small group). The number needs to be adjusted, and more recent references should/could be used.

- Page 3, lines 72-74: There is a previous paper on blood transcriptional profiles in children with symptomatic and asymptomatic cCMV and controls (Rovito et al. Plos One 2018). Although this study only included small groups and therefore had an exploratory nature, it should be mentioned here (and in the discussion). Actually some of the results on overexpressed genes in symptomatic cCMV were also detected to be overexpressed in the paper of Rovito. (LAG3, RSAD2, IFIT1, IFIT3, GZMH, OAS3)

Results:

- Patient characteristics (page 6-7): Do the authors have any information on the CMV viral load at birth in the cCMV infants? Although there is doubt whether this correlates with outcome, Rovito et al showed that some genes were overexpressed especially in newborns with high CMV viral load.

- Patient characteristics (page 6-7): to complete the patient characteristics in this paragraph, I would prefer that the outcome of the newborn hearing screen and the late onset SNHL are (also) mentioned here (now on page 11).

- Page 10-11, Longitudinal analysis: Were the expression profiles of the follow-up samples of cCMV children compared to age-matched healthy controls (so also follow-up samples of the control-group)? Or compared to the samples of the original control-group aged 11 days (mean). I could not find this information in the Methods section, so please mention this in the Methods and the legends of fig 5A/5B and 6A-B

- Page 11, lines 257-258: Please indicate the characteristics of the SNHL, unilateral or bilateral,

severity.

- Page 11, line 263: put MIR125B2 in brackets behind Mir-10.
 - Page 11, expression of the 15 classifier genes: was this compared to the healthy control group?
- Tables and figures:

- In the title of Figure 6 is "figure 6 figure 5": delete figure 5
- Supplementary table 3: Insert "Unique" before Asymptomatic cCMV gene list.
- Supplementary figures: It would be more clear if these figures are called "Supplementary Figure 1" etc.
- In suppl figures 3-5: the modular maps legend (D) is difficult to read

Discussion:

- The authors may want to reflect on the data of Rovito in their discussion.
- Page 14, lines 319-321: The interpretation of the follow-up expression profiles, and specifically the longitudinal modular analyses as shown in Figure 6, may be influenced by the choice of control group. Preferably a control group with similar follow-up samples is used, in order to correct for changes in expression profiles in the non-infected children with age. It is not clear to me whether this has been done. The authors may want to comment in their discussion.
- One of the remarkable outcomes that should be mentioned in the discussion is the fact that half of the asymptomatic children develop SNHL, which is much higher than expected. In general one could say that the study cohort is, due to the targeted CMV screening, a selected group with a relatively severe outcome, for example with 49 symptomatic children and 31 asymptomatic. And many asymptomatic children with SNHL. The group does not reflect the "normal" distribution as found in universal CMV screening. This could have influenced the results in the expression profiles as well and should therefore be mentioned.
- Page 15, line 351: SNHL instead of SNH.

Reviewer #2 (Remarks to the Author):

Overall, this is a well-design study comparing healthy, asymptomatic and symptomatic cCMV-infected infants, as well as, the incorporation of longitudinal clinical data. The use of both a the training set and validation set for the gene expression biosignatures is appreciated; furthermore, the modular gene analyses were particularly informative. Yet, the MDTH metric was less useful and needs further clarification. Finally, the 15-gene classifier that could distinguish between late SNHL+ and SNHL- children is intriguing, but it's ability to predict disease development is overstated and necessitates a validation set to make conclusions about its utility as a clinical biomarker. Moreover, there is not validation at the protein level or through a distinct RNA measure of the distinct transcriptome findings.

specific comments below:

1. Questionable utility of the "most abnormal value" section in Table 1, would consider removing as does not add useful information in my opinion
2. Would recommend adding the demographics for the healthy control group to Table 1 in addition to the symptomatic/asymptomatic cCMV-infected children (not just putting this in sup Table 1)
3. Can the authors further explain their rationale for the methodology of applying the asymptomatic/symptomatic cCMV biosignatures to the entire cohort that included the "training set" of asymptomatic/symptomatic cCMV cases respectively, would be better to consistently keep the training sets and validation sets separate
4. Please elaborate on the calculation of molecular disease to health (MDTH) score in methods and results sections (paragraph starting at line 222); in the article referenced, this genomic score is determined first with a training set and then with a validation set, do you also do the analysis this way in your study? Also, the referenced article does not mention a "molecular disease to health"

specifically, so please mention that this is a new term/analytical approach. The article mentioned (citation 12) uses a "pre-determined baseline" to calculate this metric, can you describe how you determine this baseline for your study? As far as I can tell, you do not state the analytical application of this score in your cohort nor elaborate on the importance of this score in terms of clinical utility in the manuscript, please expand upon this further. It seems that the MDTH score is most interesting in that the MDTH score is similar between symptomatic and asymptomatic groups suggesting comparable transcriptional signatures (and degree of fold-change); however, since it is generated using the "differentially" expressed transcripts between CMV-infected and healthy controls, the comparison between the infected and healthy groups is not meaningful.

5. Please explain why in Figure 4 all the control patients have a MDTH score of 0 (or nearly 0), are the comparisons all normalized against this "baseline"?

6. In lines 249-250, a significant difference between the inflammation-related genes is mentioned but the direction of expression (i.e. overexpressed or underexpressed comparatively between asymptomatic and symptomatic groups is not stated), please state this explicitly and provide beta estimate or odds ratio.

7. In lines 260-262, you mention the 15 classifier genes that predict SNHL with 100% accuracy, but don't elaborate on the expression profile of these genes – please expand upon the specifics of how this is "predictive" of hearing loss versus simply descriptive in being able to discriminate between the two groups (those that develop SNHL and not by clustering). The "predictive" capability of this 15-gene classifier was mentioned again later in the results and throughout the discussion, but as far as is clear from the manuscript it has not been able to predict disease development but simply can discriminate between SNHL+ and SNHL- cases in their small cohort (n=23), more emphasis of the necessity of analyzing this biosignature in a replication/validation cohort or prospective cohort is needed.

7. please discuss the reasons for the lack of protein-level data and downside of only including mRNA/transcript data in the study in the discussion.

8. please clarify the difference (%) on the Y-axis in the legend of figure 6 (also labeled figure 5) concerning longitudinal analysis of modular gene expression profiles.

9. in the legend of figure 7, you mention again that this 15-gene classifier is "predictive" of SNHL; however, the heatmap and PCA shown only demonstrate that these 15-genes can be used to discriminate between those with and without SNHL loss development. Since there was no training set and subsequent validation set (understandably due to the small sample size), how can you conclude that this biosignature is predictive? Please explain.

Reviewer #3 (Remarks to the Author):

I have a number of concerns with/points of clarification on the statistical analyses performed, the study design and the study population.

1. It is reported that 21 healthy age, gender and race-matched controls were enrolled at well-child visits or while undergoing minor elective surgical procedures. Are those enrolled while undergoing minor elective surgical procedures truly representative of healthy controls? When were these subjects' blood samples taken? Were they taken before or after surgery, before or after receiving some form of medication, etc.? It is hard to judge from what is described in the manuscript whether they are truly representative.

2. The authors should clarify whether or not the infants with cCMV infection (either clinically

apparent or asymptomatic) were receiving antiviral treatment at baseline or during follow-up, as presumably use of antiviral treatment can affect blood transcriptional profiles.

3. The authors should also clarify how many of the the symptomatic and asymptomatic infants had normal/abnormal hearing at birth/baseline for both the 87 cCMV infants recruited and for the 80 cCMV infants actually analysed.

4. There is a differential loss of infants to analyse in the healthy controls (11/21) versus the cCMV infected infants (7/87 - how are these 7 split between symptomatic and asymptomatic?). The authors should discuss how loss of these infants' samples can impact on the results of their analyses and the representativeness of their population.

5. It is not clear from the manuscript at what level did the authors control the false discovery rate in their supervised analyses using Mann-Whitney test. In the description on p. 6 the authors write "Supervised analysis ... was performed using a Mann-Whitney test ($P < 0.01$), followed by Benjamini-Hochberg multiple test corrections...". No way do they state at what level they are controlling the false discoveries.

6. I am not satisfied with their strategy for validation. It does not seem entirely satisfactory to just perform unsupervised (hierarchical) clustering on the test sample and same healthy controls and demonstrate that, for example, 21 of the 24 symptomatic infants group together in one cluster and the remaining (3symptomatics+10controls) fall in another. Firstly, this does not demonstrate that all 2318 genes are required to define clusters. Potentially, a much smaller subset of these transcripts could have performed equally well. Additionally, not all these 2318 genes would satisfy your supervised analysis criteria if applied to the test samples and controls. Moreover, they could have been other transcripts not identified as differentially expressed in the training analysis that could have provided a better clustering on the testing data. Equally, if the testing sample of symptomatic infants was used instead for training then you could have ended up with an entirely different transcriptional signature. Also using the same controls in the training set does not constitute an "independent cohort of 24 symptomatics CMV-infected infants and healthy controls" for validation. The above comments also apply to the development and validation of the Asymptomatic cCMV biosignature.

7. Figure 1. It would be extremely helpful if the authors ordered the genes (y-axis) and control samples (x-axis) in 1a and 1b and in 1c and 1d the same, especially as it allows an easier comparison across figures.

8. It would also be scientifically interesting to compare the asymptomatic infants with the symptomatic infants directly to see if there are transcripts which discriminate between these two groups.

9. The authors may want to consider alternative supervised approaches to identify differential expression over the Mann-Whitney test. For example, multinomial logistic regression, ANOVA, penalised regression models, support vector machines etc to build a multi-transcript classifier. And consider internal cross-validation as well as external validation.

10. It is important to stress that not statistical significant findings (i.e. $p > 0.05$) does not mean that there is no difference (or that results are "similar"). It means that there is no evidence to declare a difference. This could be entirely because of lack of power to detect the difference especially when sample sizes are small or it could be that there is truly no difference. The authors should be careful how they interpret their findings (e.g. wrt Modular Analyses). Also statistical significance does not necessarily equate to clinical significance.

11. Supplementary Table 4: What does the negative numbers in the 4th and 6th columns mean? What test is being performed to obtained the p-values?

12. Why was the analysis to develop a classifier for predicting late-onset SNHL in the asymptomatic cCMV infants restricted to only the 23 infants with a minimum of 900 days evaluation? Surely, using the other asymptomatic infants who were followed up less than 900 days would also be valuable for developing the classifier. The random forest approach for a binary outcome could be replaced, for example, by the random forest approach for a survival outcome.

13. The authors should explain why they did not consider clinical variables when developing the classifier for predicting late-onset SNHL. It could be that the 15-gene signature derived adds little to predicting late-onset SNHL in the presence of easily collected clinical variables.

14. Also would it not be of interest to develop a predictive model for late-onset SNHL in all patients infected with cCMV? Why was this not considered?

15. P 6, lines 140-142. Please re-write these sentences as they are not correct. For example, the non-parametric tests are not being used to compare between continuous variables, they are being used to compare whether there are differences in a continuous variable between groups.

15. P 13, line 295: Should "able" be "unable"

REVIEWER #1

This manuscript describes the blood genome expression profiles in neonatal whole blood in children with congenital CMV infection, and compares the profiles of asymptomatic cCMV children as well as symptomatic cCMV children with a healthy control group. Furthermore, both cCMV groups are compared to each other. This manuscript is well written, the way the experiments are performed (with a training and a test panel) is thorough and leads to robust conclusions in this specific cohort. The conclusions, that the expression profiles in asymptomatic and symptomatic children are comparable and non-distinguishable is very interesting. The information on functional genes gives insight into pathogenetic mechanisms and the authors have identified a set of classifier genes that could potentially predict which infants within the asymptomatic group have increased risk of developing late-onset hearing loss. If this can be confirmed in another cohort, this will indeed impact the clinical policy in these children.

I have to mention that I am not an expert on statistical analyses of gene expression data. I do think the authors are clear and transparent in explaining their statistical methods.

There are three points that need further attention:

Comment #1

There is a previous paper on expression profiles in neonatal whole blood on filter paper in children with cCMV and a healthy control group. This should be included in this manuscript.

Response #1. We apologize for the oversight. We now mention the manuscript published by Rovito et al. in PLoS One 2018¹ in the introduction, and discussion (page 3, line 64; page 16 lines 354-366). Please see also response #5 and #16.

Comment #2

The follow-up samples are very interesting, also in view of the longer term pathogenesis. However, it is not clear to me, whether the expression profiles at the different time points were compared to the neonatal healthy control group, or to follow-up samples of the healthy control group at similar time points. This may impact the results and conclusion.

Response #2. We apologize for the lack of clarity. When evaluating the longitudinal samples, the biosignatures derived for the symptomatic and asymptomatic cCMV cohorts were generated using the initial samples collected in the first 3 weeks of life, and the neonatal healthy control cohort. These signatures were then applied to follow up samples. Essentially, we evaluated the genes differentially expressed identified during the neonatal period to the longitudinal samples, thus allowing us to map longitudinal changes in initial abnormal gene expression. We have elaborated this further in the methods (page 6, lines 131-33), results (page 12, lines 264-267) and discussion (page 15 lines 347-349) sections of the manuscript. Please see response to #8 and #17.

Comment #3

The authors should acknowledge in the discussion that their cohort reflects a more severely affected cohort than you would expect to find through universal screening. This is important because it may influence the applicability of e.g. the classifier-gene set.

Response #3. This is a pertinent comment and appreciated. We agree that the relative severity of our cohort is greater than what would be identified via universal screening (as evidenced by the higher rate of symptomatic cCMV infection than otherwise expected). On the other hand, asymptomatic cCMV infants were identified by universal screening as part of the CHIMES study, which might be more generalizable. Details of the study cohort are included in material and methods (page 4, lines 89-91), and in supplementary material. In addition, we have now included under the discussion this study limitation (page 17, lines 384-388).

Comments:

INTRODUCTION:

Comment #4

Page 3, lines 60-67: The percentage of children that develops SNHL in the symptomatic cCMV is not 50 %, but somewhat lower. The systematic review of Goderis et al. (2014) states 1 in 3 symptomatic children will experience SNHL. The group of Fowler showed 40 % SNHL in symptomatic children and in the paper of Foulon et al. (2019) 44 % had SNHL (in a small group). The number needs to be adjusted, and more recent references should/could be used.

Response #4: We appreciate the comment, the reviewer is right. We have now adjusted the reported numbers and references accordingly (page 3, line 55). It reads as:

“Fetal infection with CMV results in sensorineural hearing loss (SNHL) in approximately 33% to 44% of infants with clinically apparent (“symptomatic”) disease....²⁻⁵.

Comment #5

Page 3, lines 72-74: There is a previous paper on blood transcriptional profiles in children with symptomatic and asymptomatic cCMV and controls (Rovito et al. Plos One 2018). Although this study only included small groups and therefore had an exploratory nature, it should be mentioned here (and in the discussion). Actually, some of the results on overexpressed genes in symptomatic cCMV were also detected to be overexpressed in the paper of Rovito. (LAG3, RSAD2, IFIT1, IFIT3, GZMH, OAS3).

Response #5: We appreciate the comment. Now we refer to this paper in both the introduction (page 3, line 64) and the discussion sections (page 16, lines 354-366). It reads as:

“Analysis of blood transcriptional profiles has provided significant insights into various infectious diseases in children, including those with cCMV.¹

“In a recent study Rovito et al analyzed blood transcriptional profiles that were derived from blood dried blood spots in 12 infants with cCMV infection and 6 healthy controls. Although, no significant differences in gene expression were identified between groups, likely related to the small sample size, LAG3, IFIT1, OAS3 or GZMH, were overexpressed in their study, and also among our patients with cCMV infection, further validating our findings.¹ “...No overlap was noted between the 16-gene classifier set and those identified by Rovito et al, which may be explained by differences in characteristics among cohorts”.

RESULTS:

Comment #6

Patient characteristics (page 6-7): Do the authors have any information on the CMV viral load at birth in the cCMV infants? Although there is doubt whether this correlates with outcome, Rovito et al showed that some genes were overexpressed especially in newborns with high CMV viral load.

Response #6: Unfortunately, quantitative blood PCR was not performed routinely in this cohort of patients. Blood PCR was performed in 25 of the 80 cCMV infants, with detectable CMV DNA in 6 (24%) patients. Only 1 of those 6 patients had a quantitative blood PCR performed (CMV loads 12,668 copies/mL). As such, we were unable to draw definitive conclusions. Nevertheless, we have now included this information in the study results (page 7, lines 156-158), and limitations (page 17, lines 377-78). It reads as:

“Blood PCR was performed in 31% (25/80) of patients, and CMV DNA detected in 6 (24%) infants. Only one of those 6 infants had a quantitative rt-PCR performed (12,668 CMV copies/mL)”.

“Quantitative blood CMV PCR was not performed routinely, and thus we are unable to compute correlations between CMV loads and transcriptional data”.

Comment #7

Patient characteristics (page 6-7): to complete the patient characteristics in this paragraph, I would prefer that the outcome of the newborn hearing screen and the late onset SNHL are (also) mentioned here (now on page 11).

Response #7: We have included this information as requested in the patient characteristics section (page 7, lines 159-163). It reads as:

“No infant with asymptomatic cCMV infection had evidence of SNHL at birth, compared with 9 (11%) infants with symptomatic cCMV infection. During the 3-year longitudinal follow-up period, 13 (27%) infants with symptomatic cCMV and 11 (35%) with asymptomatic cCMV infection developed late-onset SNHL.”

Comment #8

Page 10-11, Longitudinal analysis: Were the expression profiles of the follow-up samples of cCMV children compared to age-matched healthy controls (so also follow-up samples of the control-group)? Or compared to the samples of the original control-group aged 11 days (mean). I could not find this information in the Methods section, so please mention this in the Methods and the legends of fig 5A/5B and 6A-B

Response #8: For the longitudinal analyses, the biosignatures derived for baseline comparisons (infants with symptomatic and asymptomatic cCMV at diagnosis vs healthy age-matched controls) were applied to follow up samples on year 1, 2 and 3. We essentially evaluated the genes differentially expressed in the neonatal period over time, thus allowing us to map the longitudinal changes of the gene expression patterns identified initially. We have included this information in the methods, results sections and legends as suggested (page 6, lines 131-133; page 12, lines 264-267).

“To evaluate whether gene expression profiles evolved over time, biosignatures derived for infants with symptomatic and asymptomatic cCMV infection at baseline were applied across the longitudinal patient samples obtained on year 1, 2 and 3.”

“To be able to map changes in initial gene expression over time, the symptomatic (2,592 transcripts; Fig 5A) and asymptomatic (3,324 transcripts; Fig 5B) cCMV biosignatures were applied to the longitudinal samples within each cohort, that showed that the initial changes in overexpression of transcripts persisted up to three years of age”.

Comment #9

Page 11, lines 257-258: Please indicate the characteristics of the SNHL, unilateral or bilateral, severity.

Response #9: We have included this information in the body of the text as requested (pages 13, lines 289-293). Briefly, 52 infants with cCMV had normal hearing evaluations at baseline, of those 24 infants (11 with asymptomatic cCMV and 13 with symptomatic cCMV) developed SNHL at follow-up.

- Of the 11 infants in the asymptomatic cCMV cohort that developed SNHL, 7 were bilateral, and 4 unilateral, and of mild (n=7) or moderate (n=3) severity. The severity in one patient was not reported.
- Of the 13 infants with symptomatic cCMV infection that developed SNHL, 8 were bilateral and 5 unilateral, 11 mild and 2 severe.

Comment #10

Page 11, line 263: put MIR125B2 in brackets behind Mir-10.

Response #10: Because of the adjustments performed in our analyses (as requested by reviewer 3), this gene was no longer identified in our set, though we appreciate the recommendation of the reviewer.

Comment #11

Page 11, expression of the 15 classifier genes: was this compared to the healthy control group?

Response #11: Healthy controls were not included in the prediction models. As suggested by reviewer #3 we have reanalyzed the data and we now include all infants with cCMV (symptomatic or asymptomatic) who developed SNHL (n=24) and compared them to infants with cCMV who did not developed SNHL (n=28). Median expression values of the 16 classifier genes identified are displayed in Fig 7. We have clarified this aspect in the results section (page 13, lines 294-303) and Figure 7 legend. It reads as:

“Random Forest-Recursive Feature Elimination (RF-RFE) analyses applied to samples obtained at the time of cCMV diagnosis in these 52 cCMV infants (24 who developed late-onset SNHL and 28 who did not), identified 16 classifier genes (Supplementary Table S8) that were associated with late-onset SNHL with 92% accuracy, and an area under the curve (AUC) of 0.97 (Supplementary Fig S7). Median expression values for those genes in both groups of infants with or without late onset SNHL are displayed in Figure 7A”.

TABLES AND FIGURES:

Comment #12

In the title of Figure 6 is “figure 6 figure 5”: delete figure 5

Response #12: Thank you for identifying this error – we have made the adjustments as mentioned.

Comment #13

Supplementary table 3: Insert “Unique” before Asymptomatic cCMV gene list.

Response #13: We have included this clarification as requested.

Comment #14

Supplementary figures: It would be more clear if these figures are called “Supplementary Figure 1” etc.

Response #14: We have made those adjustments for clarification as requested and now all supplementary material is easier to recognize.

Comment #15

In suppl figures 3-5: the modular maps legend (D) is difficult to read

Response #15: We appreciate the comment. We have created and included a new modular map legend that we believe it is easier to read.

DISCUSSION:

Comment #16

The authors may want to reflect on the data of Rovito in their discussion.

Response #16: We have added additional comments regarding the Rovito’s manuscript and how it relates to our results to the discussion section. Please see response to comment #5 (page 16, lines 354-365).

Comment #17

Page 14, lines 319-321: The interpretation of the follow-up expression profiles, and specifically the longitudinal modular analyses as shown in Figure 6, may be influenced by the choice of control group. Preferably a control group with similar follow-up samples is used, in order to correct for changes in expression profiles in the non-infected children with age. It is not clear to me whether this has been done. The authors may want to comment in their discussion.

Response #17: We appreciate this comment. When evaluating the longitudinal samples, the biosignatures from infants with symptomatic and asymptomatic cCMV infection were generated using the initial samples from cCMV patients obtained at the time of diagnosis, and the neonatal age-matched healthy control cohort. These signatures were then applied to follow up longitudinal samples, obtained from symptomatic and asymptomatic cCMV infected children, essentially evaluating the expression of the same genes identified in the neonatal period over time.

By using the initial infant samples as the reference point for the longitudinal analysis, we were able to map the longitudinal changes of the initial abnormal gene expression identified in the neonatal period, in each individual patient over time. We have clarified the reasoning behind these analyses further in the methods (*page 6, lines 131-33*), results (*page 12, lines 264-67*) and discussion (*page 15 lines 347-49*) sections of the manuscript. It reads as:

“To evaluate whether gene expression profiles identified in the neonatal period evolved over time, the biosignatures derived for infants with symptomatic and asymptomatic cCMV infection at baseline were applied to the longitudinal patient samples obtained on years 1, 2 and 3”.

“To be able to map changes in initial gene expression over time, the symptomatic (2,592 transcripts; Fig 5A) and asymptomatic (3,324 transcripts; Fig 5B) cCMV biosignatures were applied to the longitudinal samples within each cohort, that showed that the initial changes in overexpression of transcripts persisted up to three years of age”.

“...when we applied the biosignatures of symptomatic and asymptomatic cCMV infection to longitudinal samples, the abnormal immune profiles identified during the first three weeks of life persisted for years after initial testing”.

Comment #18

One of the remarkable outcomes that should be mentioned in the discussion is the fact that half of the asymptomatic children develop SNHL, which is much higher than expected. In general, one could say that the study cohort is, due to the targeted CMV screening, a selected group with a relatively severe outcome, for example with 49 symptomatic children and 31 asymptomatic. And many asymptomatic children with SNHL. The group does not reflect the “normal” distribution as found in universal CMV screening. This could have influenced the results in the expression profiles as well and should therefore be mentioned.

Response #18: We appreciate the reviewer’s comment, and agree that within our cohort we had a high rate of abnormal hearing evaluations in the asymptomatic cCMV cohort. Although asymptomatic infants were originally identified by universal screening in the CMV & Hearing Multicenter Screening (CHIMES) study, they may have represented a more biased population, and findings would need to be validated in a new infant cohort with cCMV. We have now addressed this limitation in the discussion section (*page 17, lines 386-388*). It reads as:

“...the cohort of asymptomatic cCMV infants enrolled developed SNHL at higher rate than previously reported,³ and thus may reflect a biased population with greater disease severity, thus limiting the generalizability”.

Comment #19

Page 15, line 351: SNHL instead of SNH.

Response #19: Thank you for identifying this error – we have made the adjustments as mentioned.

REVIEWER #2

Overall, this is a well-design study comparing healthy, asymptomatic and symptomatic cCMV-infected infants, as well as, the incorporation of longitudinal clinical data. The use of both the training set and validation set for the gene expression biosignatures is appreciated; furthermore, the modular gene analyses were particularly informative. Yet, the MDTH metric was less useful and needs further clarification. Finally, the 15-gene classifier that could distinguish between late SNHL+ and SNHL- children is intriguing, but it's ability to predict disease development is overstated and necessitates a validation set to make conclusions about its utility as a clinical biomarker. Moreover, there is not validation at the protein level or through a distinct RNA measure of the distinct transcriptome findings.

Specific comments below:

Comment #1

Questionable utility of the “most abnormal value” section in Table 1, would consider removing as does not add useful information in my opinion.

Response #1: We appreciate the comment. We have now removed the “most abnormal value” in the table (now Table 2).

Comment #2

Would recommend adding the demographics for the healthy control group to Table 1 in addition to the symptomatic/asymptomatic cCMV-infected children (not just putting this in sup Table 1)

Response #2: We appreciate reviewers' comments. We have now included a new table (Table 1) with the demographic characteristics of the healthy control group and of all infants with symptomatic (training and test) and asymptomatic (training and test) cCMV infection. For clarity Table 1 is separate from Table 2, as the healthy cohort group lacked several of the evaluation work-up performed in infants with congenital CMV. Additional information of the demographic data for the training sets for each cohort (symptomatic and asymptomatic), is now included in Supplementary Table 3 and showed no significant differences in any of the parameters analyzed.

Comment #3

Can the authors further explain their rationale for the methodology of applying the asymptomatic/symptomatic cCMV biosignatures to the entire cohort that included the “training set” of asymptomatic/symptomatic cCMV cases respectively, would be better to consistently keep the training sets and validation sets separate

Response #3: Once we defined the signatures of symptomatic and asymptomatic cCMV infection, we hypothesized that infants with symptomatic cCMV disease had a pattern of unique differential gene expression compared with infants with asymptomatic infection. Therefore, one would expect discrete clustering among the groups (healthy controls, symptomatic cCMV, and asymptomatic cCMV) if we apply the symptomatic cCMV biosignature to the entire cohort. Surprisingly, we did not observe any distinct clustering between symptomatic and asymptomatic cCMV infants when we performed this unsupervised analysis. To verify these findings, we performed the exact same unsupervised analysis with the asymptomatic cCMV biosignature and found similar results. These analyses (as well as modular analyses and MDTH scoring) revealed striking similarities among infants with cCMV infection, irrespective of whether they had symptomatic or asymptomatic disease. We have now elaborated further the rationale for conducting these analyses in the results section (*page 10, lines 224-226*). It reads as:

“...thus, while in both analyses all ten healthy control infants clustered separately from the cCMV-infected infants, neither the asymptomatic or symptomatic cCMV biosignatures were able to reliably distinguish cCMV infected infants based on their clinical classification.”

Comment# 4.

Please elaborate on the calculation of molecular disease to health (MDTH) score in methods and results sections (paragraph starting at line 222); in the article referenced, this genomic score is determined first with a training set and then with a validation set, do you also do the analysis this way in your study? Also, the referenced article does not mention a “molecular disease to health” specifically, so please mention that this is a new term/analytical approach. The article mentioned (citation 12) uses a “pre-determined baseline” to calculate this metric, can you describe how you determine this baseline for your study? As far as I can tell, you do not state the analytical application of this score in your cohort nor elaborate on the importance of this score in terms of clinical utility in the manuscript, please expand upon this further. It seems that the MDTH score is most interesting in that the MDTH score is similar between symptomatic and asymptomatic groups suggesting comparable transcriptional signatures (and degree of fold-change); however, since it is generated using the “differentially” expressed transcripts between CMV-infected and healthy controls, the comparison between the infected and healthy groups is not meaningful.

Comment #5:

Please explain why in Figure 4 all the control patients have a MDTH score of 0 (or nearly 0), are the comparisons all normalized against this “baseline”?

Responses #4&5: We apologize for the lack of clarity and certainly agree with the reviewer that the main point is that MDTH scores did not differentiate between symptomatic and asymptomatic cCMV infection. Briefly, the MDTH scores convert the global transcriptional perturbation of each patient sample into a numeric value, or score, that can be incorporated into analyses of clinical variables. To calculate MDTH scores median expression values (and fold-change) of all significantly differentially over and underexpressed genes from each individual cCMV patient, either symptomatic or asymptomatic, were compared with the median expression values of healthy controls, that are used as a reference. Healthy controls thus served as the predetermined baseline explaining why their score is quite low (median MDTH score: 44.07 [IQR 12.8-106.3]) relative to infants with symptomatic (1704 [1067-2021]) or asymptomatic (1522 [1001-2284]) cCMV infection.

The MDTH score concept was first described by our collaborator Dr. Chaussabel in the manuscript pointed out by the reviewer (*Pankla et al, Genome Biol 2009*)⁶. Since then, our group have used this metric in different studies to help with patient classification and assessment of disease severity.⁷⁻¹³ We have now clarified further the importance and meaning of the MDTH genomic score in the methods (*page 6, lines 126-130*) and results sections (*page 11, lines 251-257*). It reads as:

“Molecular distance to health (MDTH), a tool that converts the global transcriptional perturbation of each individual patient sample into an objective score in relation to the healthy control baseline, was calculated and compared between symptomatic and asymptomatic infants with cCMV infection and healthy controls”.

“To investigate whether global transcriptional differences allowed discrimination between infants with symptomatic and asymptomatic cCMV infection, we calculated the molecular distance to health (MDTH) genomic score. This metric summarizes into a numeric value the global transcriptional perturbation of each individual patient sample compared to age-matched healthy controls. To calculate the MDTH scores, 3,756 transcripts identified in either the symptomatic or asymptomatic cCMV biosignatures were utilized (Supplementary Fig S3).”

Comment #6

In lines 249-250, a significant difference between the inflammation-related genes is mentioned but the direction of expression (i.e. overexpressed or underexpressed comparatively between asymptomatic and symptomatic groups is not stated), please state this explicitly and provide beta estimate or odds ratio.

Response #6: We apologize for the lack of clarity. Inflammation genes were initially underexpressed in both symptomatic and asymptomatic cCMV infection, their expression levels normalized in the first year of life and then were significantly overexpressed in children with asymptomatic vs symptomatic cCMV on the second year of life. The data plotted represents the mean difference of the percentages of genes upregulated and downregulated within a given module. Therefore, a positive % difference means that the proportion of upregulated genes is higher than downregulated genes within a module, whereas a negative % difference represents more downregulated genes than upregulated genes within a module. We have added additional clarification to the results section and figure legend to address this comment (*page 12, lines 270-272*). Please also see response to comment #9. It reads as:

“Modular analyses were also applied to the longitudinal samples and revealed that while initial inflammation transcripts were underexpressed in infants with symptomatic and asymptomatic cCMV infection, expression levels normalized in year one and were significantly overexpressed in infants with asymptomatic vs symptomatic cCMV infection ($p < 0.009$) at 2 years of age”.

“Figure 6: Longitudinal modular analyses of infants with symptomatic and asymptomatic cCMV infection. Mean percent difference in modular expression of innate (inflammation, neutrophil, monocytes, interferon) and adaptive immunity (T-cell and B-cell/plasma cell) modules are plotted and compared over time (baseline, one year, two years and three years of age) for infants with symptomatic (orange) and asymptomatic (green) congenital CMV infection. *The mean percent difference was calculated by the percentage of genes upregulated (positive percent) minus the percentage of downregulated genes (negative percent) within each module. When the biological functions were represented by more than one module (i.e. interferon, inflammation, T-cells, see Figure 3), modular mean was derived and the percentage difference calculated as explained above”.*

Comment #7

In lines 260-262, you mention the 15 classifier genes that predict SNHL with 100% accuracy, but don't elaborate on the expression profile of these genes – please expand upon the specifics of how this is “predictive” of hearing loss versus simply descriptive in being able to discriminate between the two groups (those that develop SNHL and not by clustering) the “predictive” capability of this 15-gene classifier was mentioned again later in the results and throughout the discussion, but as far as is clear from the manuscript it has not been able to predict disease development but simply can discriminate between SNHL+ and SNHL- cases in their small cohort (n=23), more emphasis of the necessity of analyzing this biosignature in a replication/validation cohort or prospective cohort is needed

Response #7: We appreciate the comment. To the reviewers' point, one is correct in stating that the classifier genes do discriminate between infants who develop SNHL and those who do not. We used the word “predictive” to indicate the timing for which samples were obtained, as the classifier analyses were conducted with samples obtained at first enrollment (within the first three weeks of life) and before knowing which infants would go onto developing SNHL in subsequent years. We have now reworded the message of the manuscript and have toned down our statements as we agree on the need for validation of our findings in independent cohorts (*page 2, lines 48-51, page 16, lines 359-61 & 365-366, page 17, lines 386-388*).

“By applying a Random Forest classification algorithm to infants with cCMV infection and normal hearing in the newborn period, we identified a group of 16 genes that were associated with the development of late-onset SNHL with 92% accuracy”. “While these data is encouraging, validation in larger cohorts of patients with adequate follow-up is necessary”.

“Despite these limitations, and the need for validation in independent cohorts...”

Comment #8

Please discuss the reasons for the lack of protein-level data and downside of only including mRNA/transcript data in the study in the discussion

Response #8: Unfortunately, due to blood-volume limitations, protein level data was not performed (similar to the lack of functional assays to correlate our immune profiles). This is a limitation that has since been noted in the discussion section (page 17, lines 378-381).

“Similarly, because of limitations in blood samples volume, we were not able to validate the data at the protein level or to perform functional assays to correlate transcriptional profiles and functional immune responses, and those studies should be conducted in the future”.

Comment #9

Please clarify the difference (%) on the Y-axis in the legend of figure 6 (also labeled figure 5) concerning longitudinal analysis of modular gene expression profiles.

Comment #9: Thank you for noting the additional figure annotation which has been corrected. The % difference reflects the mean difference of the percentages of genes upregulated and downregulated within a given module. Therefore, a positive % difference means that the proportion of upregulated genes is higher than downregulated genes within a module, whereas a negative % difference represents more downregulated genes than upregulated genes within a module. Some of the biological functions are represented by more than one module (ex. interferon, inflammation, T-cells, see Figure 3). In those cases, a modular mean was derived and the percentage difference calculated as explained above. All these aspects have now been clarified under the Figure legend. This has been clarified under the figure legend (see response to #6).

Comment #10

In the legend of figure 7, you mention again that this 15-gene classifier is “predictive” of SNHL; however, the heatmap and PCA shown only demonstrate that these 15-genes can be used to discriminate between those with and without SNHL loss development. Since there was no training set and subsequent validation set (understandably due to the small sample size), how can you conclude that this biosignature is predictive? Please explain.

Response #10. We agree with the reviewer, please see response to #7. We have toned down our statements as our findings will need further validation.

REVIEWER #3

I have a number of concerns with/points of clarification on the statistical analyses performed, the study design and the study population.

Comment #1

It is reported that 21 healthy age, gender and race-matched controls were enrolled at well-child visits or while undergoing minor elective surgical procedures. Are those enrolled while undergoing minor elective surgical procedures truly representative of healthy controls? When were these subjects blood samples taken? Were they taken before or after surgery, before or after receiving some form of medication, etc.? It is hard to judge from what is described in the manuscript whether they are truly representative.

Response #1: We apologize for the lack of clarity regarding the enrollment of healthy controls. As the reviewer probably agrees, obtaining blood samples from healthy infants is always challenging, specially at that young age. One of the strategies that we have implemented to obtain blood samples from otherwise healthy young infants, is to enroll them while undergoing minor elective surgical procedures such as a hernia repair, ear tag removal or circumcision (given the age of the infants). We have used this approach for different studies with success^{7-10, 14-16}.

We are very strict with our exclusion criteria which include: presence of any comorbidity, immunocompromised state, use of systemic steroids or antibiotics, and an acute illness event within two weeks of enrollment. In infants enrolled while undergoing minor surgical procedures, blood samples were obtained when the intravenous (IV) line was placed for induction and medication administration, and before the actual procedure was performed, so the research team would not interfere with the personnel at the operating room. We have now clarified this aspect in the material and methods section (*page 4, lines 80-84*). It reads as:

“In parallel, we enrolled a cohort of healthy age, gender-, and race-matched controls at well-child visits or prior to undergoing minor elective surgical procedures. Healthy controls were excluded from the study if they had an acute illness or exposure to antibiotics or steroids within 2-weeks of enrollment, or any underlying comorbidity”.

Comment #2

The authors should clarify whether or not the infants with cCMV infection (either clinically apparent or asymptomatic) were receiving antiviral treatment at baseline or during follow-up, as presumably use of antiviral treatment can affect blood transcriptional profiles.

Response #2. That is an excellent point. None of the infants with asymptomatic cCMV received antiviral therapy. Of the infants with symptomatic cCMV infection (n=49), only two patients (NCH_913 and NCH_925) were receiving valganciclovir for 3 and 6 days respectively at the first study visit (baseline). These two patients were included in the test set and grouped with all other symptomatic cCMV patients (Fig 1B, PCA), suggesting that the effect of valganciclovir on gene expression profiles was marginal.

In addition, 39% (19/49) of infants with symptomatic cCMV infection, received antiviral therapy for either 6 weeks (9/19) or 6 months (10/19). Thus, follow-up samples at 1, 2, and 3 years of age were obtained while off antiviral therapy. This has been clarified in the manuscript (*page 8, lines 172-175*). It reads as:

“Baseline samples for transcriptome analyses were collected before initiation of antiviral therapy with the exception of 2 (3%) patients who were receiving valganciclovir for 3 and 6 days respectively at first study visit. Follow-up samples on years 1, 2 and 3 were collected off antiviral therapy”.

Comment #3

The authors should also clarify how many of the symptomatic and asymptomatic infants had normal/abnormal hearing at birth/baseline for both the 87 cCMV infants recruited and for the 80 cCMV infants actually analysed.

Response #3. By the definitions used in the study, no infant with asymptomatic cCMV infection (n=31) had an abnormal hearing evaluation at birth. With respect to infants with symptomatic cCMV, a total of 9 infants had abnormal hearing evaluation at birth. These 9 infants were included for downstream analysis within the 80 infants included in the manuscript, and thus within the original cohort of recruited patients. We have added this information to the results section (*page 7, lines 159-163*) and Table 2. It reads as:

“No infant with asymptomatic cCMV infection had SNHL at birth compared with 9 (11%) infants with symptomatic cCMV infection. During the 3-year longitudinal follow-up period, 13 (27%) infants with symptomatic cCMV and 11 (35%) with asymptomatic cCMV infection developed late-onset SNHL”.

Comment #4

There is a differential loss of infants to analyse in the healthy controls (11/21) versus the cCMV infected infants (7/87 - how are these 7 split between symptomatic and asymptomatic?). The authors should discuss how loss of these infants' samples can impact on the results of their analyses and the representativeness of their population.

Response #4. Thanks for bringing this up. We have performed further analyses to assure that the excluded patients did not significantly impacted our results. The reason behind not including those samples from both healthy controls and cCMV patients was due to insufficient or low quality RNA.

(A) With regards to the healthy control group, we compared the demographic characteristics of the controls included and not included in downstream analyses and found no significant differences between them (new Supplementary Table S1). The loss of samples brings a limitation in that we were unable to use an independent healthy control cohort for validation purposes. The main reason behind the lack of additional healthy controls for validation purposes, was in relation to the challenge of obtaining blood samples from healthy young infants. Although not ideal, we have used this strategy in the past^{10, 17}. We have added further discussion regarding this limitation to the manuscript (*page 17, lines 374-378* and Table S1). It reads as:

“The same healthy controls were used throughout all analyses and thus, an independent healthy control cohort was not included with the validation sets. While not ideal, the challenge of enrolling healthy controls (particularly young infants) has led to similar limitations in prior studies with consistent results”.^{9,10}

Supplementary Table S1. Demographic characteristics of healthy controls included and excluded from the analyses

	HC included (n=10)	HC excluded (n=11)	p-value
Age (sample collection)	11 (7-33)	21 (1-50)	> 0.99
Gestational age (weeks)	38 (35-40)	40 (39-40)	0.062
Sex (M:F)	7M:3F	5M:6F	0.39
Race (W:B)	8W:2B	7W:4B	0.64

HC: healthy controls, M; males, F: females, W: white, B: black.

(B) After careful review of the patients with cCMV infection excluded, we realized that there were 6 (instead of 7) patients not included in downstream analyses because of low quality blood RNA. The 7th patient not included had actually perinatal HSV and not cCMV. We do apologize for the oversight. Nevertheless, all 6 patients with cCMV infection not included were categorized as symptomatic cCMV infection and none had abnormal hearing at birth. We have compared the clinical characteristics of infants with symptomatic cCMV included (n=49) and not included (n=6) in downstream analyses and found that the clinical characteristics were not significantly different. This information has been included in the manuscript (*page 7, lines 148-150, and new Supplementary Table S2*). It reads as:

“The demographic and clinical characteristics between the cCMV patients and healthy controls included and not included in the study were comparable (Supplementary Tables S1 and S2)”.

Supplementary Table S2. Clinical findings of included and excluded Symptomatic CMV infants

	Symptomatic CMV Included (n=49)	Symptomatic CMV Excluded (n=6)	P value
Demographic Information			
Sex, males, n (%)	30 (61%)	2 (33%)	>0.99
Gestational age (weeks)	39 (38-40)	37.5 (36-39)	0.10
Birth weight (grams)	3,035 (2,457-3,278)	2,968 (2,199-3,190)	0.52
Birth length (cm)	48 (46-49)	47 (42.5-48.5)	0.23
Head circumference (cm)	33.4 (32-34.4)	33 (29.5-33.5)	0.22
Age at sample collection (days)	17 (11-22)	17 (9.5-23)	0.99
Examination findings, n (%)			
Rash	16 (33%)	2 (33%)	>0.99
Splenomegaly	12 (24%)	3 (50%)	0.66
Hepatomegaly	11 (22%)	2 (33%)	0.62
SGA	10 (20%)	0	0.58
Microcephaly	8 (16%)	2 (33%)	0.30
IUGR	5 (10%)	1 (17%)	0.51
Laboratory results (at diagnosis)			
WBC (cells/mm ³)	10,735 (8,355-13,133)	13,645 (10,830-16,528)	0.20
Hemoglobin (g/dL)	13.6 (11.3-17.2)	17 (13.7-18.9)	0.14
Hematocrit (%)	40.1 (33.4-48.4)	49.6 (38.9-55)	0.16
Platelet count (/mm ³)	257,000 (94,250-391,500)	290,000 (131,000-442,250)	0.80
ALT (U/L)	20 (14-31)	25 (15.5-63.5)	0.49
Direct bilirubin (mg/dL)	0.3 (0.2-0.53)	0.4 (0.2-3)	0.63
Audiologic findings, n (%)			
Abnormal ABR at any time	22 (45%)	2 (33%)	0.69
Abnormal initial ABR	9 (18%)	0	0.57
Abnormal follow up ABR	13 (32%)	2 (33%)	0.66

SGA: small for gestational age; IUGR: intrauterine growth restriction. Demographic, examination findings and laboratory results at diagnosis. Statistical analyses were performed using U-Man Whitney for non-parametric continuous variables and data reported as median, 25%-75% interquartile ranges, and the Fisher's exact or Chi-square tests for categorical data.

Comment #5

It is not clear from the manuscript at what level did the authors control the false discovery rate in their supervised analyses using Mann-Whitney test. In the description on p. 6 the authors write "Supervised analysis ... was performed using a Mann-Whitney test ($P < 0.01$), followed by Benjamini-Hochberg multiple test corrections...". No way do they state at what level they are controlling the false discoveries.

Response #5: We apologize for the oversight. The false discovery rate (FDR) was set at 1% (FDR < 0.01). We have clarified this further in the manuscript to avoid confusion.

Comment #6

I am not satisfied with their strategy for validation. It does not seem entirely satisfactory to just perform unsupervised (hierarchical) clustering on the test sample and same healthy controls and demonstrate that, for example, 21 of the 24 symptomatic infants group together in one cluster and the remaining (3symptomatics+10controls) fall in another. Firstly, this does not demonstrate that all 2318 genes are required to define clusters. Potentially, a much smaller subset of these transcripts could have performed equally well. Additionally, not all these 2318 genes would satisfy your supervised analysis criteria if applied to the test samples and controls. Moreover, they could have been other transcripts not identified as differentially expressed in the training analysis that could have provided a better clustering on the testing data. Equally, if the testing sample of symptomatic infants was used instead for training then you could have ended up with an entirely different transcriptional signature. Also using the same controls in the training set does not constitute an "independent cohort of 24 symptomatic CMV-infected infants and healthy controls" for validation. The above comments also apply to the development and validation of the Asymptomatic cCMV biosignature.

Response #6: We thank the reviewer for the comments and observations, and certainly agreed with the lack of an independent cohort of healthy controls for validation purposes (see response #4). Nevertheless, our analyses followed the standard tenets of high dimensional biological data and analysis plan.^{7, 9-12, 18-25} Specifically, differential expression analysis was performed in the derivation set (training set) to generate gene lists for the purpose of obtaining biological insight and understanding. The hierarchical clusters performed on the test sets were applied to validate the biological insights obtained from the test sets. Lastly, class prediction was performed to determine whether a parsimonious gene list was capable of predicting a clinical outcome (see response #14). We have now performed a number of analyses to validate the symptomatic and asymptomatic cCMV signatures that we hope will satisfy the reviewers concerns. This data is included in the manuscript (page 8, lines 176-179; page 9 lines 197-8 and 203-208; Fig 1, Figure S2)

1. We derived the training signatures for symptomatic and asymptomatic cCMV using linear mixed models (FDR 1%) adjusted for multiple tests correction and 1.5-fold change. These signatures were validated using principal component analyses (PCA) (new Fig 1)

2. We derived the training and test set signatures of infants with symptomatic and asymptomatic cCMV infection independently (except for the healthy controls) and correlated both signatures. For the training and test sets in the symptomatic cCMV cohort the Spearman's correlation coefficient was $r=0.88$ ($p<0.0001$), and for the training and test sets in the asymptomatic cCMV cohort the $r=0.94$ ($p<0.0001$). This information has been added to the results section (*page 9, lines 203-208*) and included as supplementary material (Supplementary Figure S2).

3. In addition to validating the signatures at the gene level, we also performed modular analyses. In these analyses, modular maps are derived from the training and test sets separately. As shown in Supplementary Figures S4&S5 modular maps for training and test sets of infants with symptomatic or asymptomatic cCMV infection were comparable. Spearman's correlation coefficient for the training and test sets in infants with symptomatic cCMV was $r=0.90$; $p<0.0001$; and between the training and test sets of asymptomatic cCMV disease $r=0.93$; $p<0.0001$.

Comment #7

Figure 1. It would be extremely helpful if the authors ordered the genes (y-axis) and control samples (x-axis) in 1a and 1b and in 1c and 1d the same, especially as it allows an easier comparison across figures.

Response #7. Following reviewer’s suggestion, to be able to maintain the order of genes exactly the same in the training and test sets, we have merged the samples for the training and test sets in the symptomatic and asymptomatic cCMV cohorts so trends in gene expression could be visualized across all samples belonging to the same condition (see Figure below). The order of controls is now exactly the same. As we have performed several new analyses to provide greater clarity (and thus generated a number of new figures), we elected not to include this figure as part of the formal manuscript, though have provided it for review.

Comment #8

It would also be scientifically interesting to compare the asymptomatic infants with the symptomatic infants directly to see if there are transcripts which discriminate between these two groups.

Response #8. Following the reviewers’ suggestion, we compared the signatures of the symptomatic and asymptomatic cCMV infants and found no significantly differentially expressed genes. We have added this information to the manuscript (*page 10, lines 215-216*) and have now included a Venn Diagram that compares the symptomatic and asymptomatic signatures and the direct comparison between symptomatic and asymptomatic cCMV infants (Supplementary Fig S3).

Comment #9

The authors may want to consider alternative supervised approaches to identify differential expression over the Mann-Whitney test. For example, multinomial logistic regression, ANOVA, penalised regression models, support vector machines etc to build a multi-transcript classifier. And consider internal cross-validation as well as external validation.

Response #9: Following reviewer's suggestions and in collaboration with Dr. Blankenship (biostatistician) listed now as co-author, and Dr. Xu (bioinformatician) both with ample expertise in the analyses of transcriptome data, we have re-analyzed all the data using linear mixed models and Bayesian methods adjusted for age (LIMMA package 3.42 in R). P-values were corrected with Benjamini-Hochberg multiple test correction test, and differentially expressed genes obtained with an FDR cut-off <0.01 and 1.5-fold-change in gene expression. This has been updated in the methods section (page 6, lines 121-123). Briefly, to derive and validate the biosignatures we used supervised and unsupervised analyses (see response #6) with the purpose of obtaining biological insight and understanding of the pathogenesis of symptomatic and asymptomatic cCMV infection. Next to determine whether a reduced number of genes were capable of predicting the development of SNHL, we used class prediction (Random Forest-Recursive elimination feature-RF-RFE-) with internal and external cross-validation (see response #13).

Comment #10

It is important to stress that not statistically significant findings (i.e. $p > 0.05$) does not mean that there is no difference (or that results are "similar"). It means that there is no evidence to declare a difference. This could be entirely because of lack of power to detect the difference especially when sample sizes are small or it could be that there is truly no difference. The authors should be careful how they interpret their findings (e.g. Modular Analyses). Also, statistical significance does not necessarily equate to clinical significance.

Response #10: We appreciate the comment and have reviewed the manuscript carefully to address these concerns.

Comment #11

Supplementary Table 4: What does the negative numbers in the 4th and 6th columns mean? What test is being performed to obtain the p-values?

Response #11: This is an oversight. We meant to indicate the direction of the expression (negative = underexpressed). We have provided greater clarification in supplementary Table S6 (former Table S4).

Comment #12

Why was the analysis to develop a classifier for predicting late-onset SNHL in the asymptomatic cCMV infants restricted to only the 23 infants with a minimum of 900 days evaluation? Surely, using the other asymptomatic infants who were followed up less than 900 days would also be valuable for developing the classifier. The random forest approach for a binary outcome could be replaced, for example, by the random forest approach for a survival outcome.

Response #12: We apologize for the lack of clarity. In effect, we used all patients with asymptomatic (and now symptomatic) cCMV infection that had normal hearing at baseline but developed SNHL at any time point during the follow-up period. We restricted however, the comparator group (infants with symptomatic and asymptomatic cCMV and no SNHL) to those infants that had at least 900 days of follow-up, due to potential for development of late onset hearing loss. After consultation with Dr. Blankenship it was suggested that the best prediction model that will fit this data was the RF-RFE algorithm (please see response #13) due to the relatively consistent timing the outcome was measured for each subject. This has been clarified in the manuscript (page 6, lines 134-37; page 13, lines 294-98, and in Supplementary Material).

Comment #13

The authors should explain why they did not consider clinical variables when developing the classifier for predicting late-onset SNHL. It could be that the 15-gene signature derived add little to predicting late-onset SNHL in the presence of easily collected clinical variables.

Response #13: As per reviewer' suggestions we have now included clinical variables in our prediction model. To identify classifier genes associated with the development of SNHL, we applied the Random Forest-Recursive Feature Elimination (RF-RFE) algorithm to select the best features for prediction modeling. Two clinical variables (neuroimaging, and microcephaly at diagnosis) were included for feature selection, which started with 3/4 of samples with 10-fold internal cross validation, and the remaining 1/4 of samples were used for external validation. The model with lowest number of features along with the best accuracy was selected for prediction. Using this strategy RF-RFE identified 16 classifier genes (AUC 0.97; accuracy 92%) associated with SNHL. See below Fig 7 and supplemental Figure S7, and response #14.

Comment #14.

Also would it not be of interest to develop a predictive model for late-onset SNHL in all patients infected with cCMV? Why was this not considered?

Response #14: Following reviewer #3 suggestions we have now developed a predictive model based on the development of subsequent SNHL but independent of the patient's clinical classification (symptomatic or asymptomatic). Of all patients enrolled 24 cCMV infants developed SNHL at some point during the follow-up period (11 asymptomatic/13 symptomatic cCMV) and 28 infants with cCMV infection who had at least 900 days of follow-up and did not develop SNHL (Supplementary Table S7). The Random Forest-Recursive Feature Elimination algorithm identified a 16 gene classifier set that was able to identify cCMV infants who developed SNHL with 92.3% accuracy (see response #13). We have updated this information under results (page 13, lines 204-298; Fig 7, and supplementary Fig S7).

Comment #15

P 6, lines 140-142. Please re-write these sentences as they are not correct. For example, the non-parametric tests are not being used to compare between continuous variables, they are being used to compare whether there are differences in a continuous variable between groups.

Response #15: The sentences have been rewritten for accuracy as requested (page 6, lines 135-137). It reads as:

“Briefly, non-parametric tests (either Mann-Whitney or Kruskal Wallis) were used to evaluate differences in continuous variables between groups, whereas differences in proportions were assessed using Fisher’s exact and chi-square test as appropriate”.

Comment #16

P 13, line 295: Should "able" be "unable"

Response #16. Reviewer #3 is correct; it should read as unable. We apologize for the error, and it has been corrected in the revision.

BIBLIOGRAPHY

1. Rovito R, Warnatz HJ, Kielbasa SM, et al. Impact of congenital cytomegalovirus infection on transcriptomes from archived dried blood spots in relation to long-term clinical outcome. *PLoS One*. 2018;13:e0200652.
2. Morton CC, Nance WE. Newborn hearing screening--a silent revolution. *The New England journal of medicine*. 2006;354:2151-2164.
3. Fowler KB, McCollister FP, Dahle AJ, Boppana S, Britt WJ, Pass RF. Progressive and fluctuating sensorineural hearing loss in children with asymptomatic congenital cytomegalovirus infection. *The Journal of pediatrics*. 1997;130:624-630.
4. Goderis J, De Leenheer E, Smets K, Van Hoecke H, Keymeulen A, Dhooge I. Hearing loss and congenital CMV infection: a systematic review. *Pediatrics*. 2014;134:972-982.
5. Foulon I, De Brucker Y, Buyl R, et al. Hearing Loss With Congenital Cytomegalovirus Infection. *Pediatrics*. 2019;144.
6. Pankla R, Buddhisa S, Berry M, et al. Genomic transcriptional profiling identifies a candidate blood biomarker signature for the diagnosis of septicemic melioidosis. *Genome Biol*. 2009;10:R127.
7. Mejias A, Dimo B, Suarez NM, et al. Whole blood gene expression profiles to assess pathogenesis and disease severity in infants with respiratory syncytial virus infection. *PLoS medicine*. 2013;10:e1001549.
8. Wallihan RG, Suarez NM, Cohen DM, et al. Molecular Distance to Health Transcriptional Score and Disease Severity in Children Hospitalized With Community-Acquired Pneumonia. *Front Cell Infect Microbiol*. 2018;8:382.
9. Jaggi P, Mejias A, Xu Z, et al. Whole blood transcriptional profiles as a prognostic tool in complete and incomplete Kawasaki Disease. *PloS one*. 2018;13:e0197858.
10. Heinonen S, Jartti T, Garcia C, et al. Rhinovirus Detection in Symptomatic and Asymptomatic Children: Value of Host Transcriptome Analysis. *Am J Respir Crit Care Med*. 2016;193:772-782.
11. Banchereau R, Jordan-Villegas A, Ardura M, et al. Host immune transcriptional profiles reflect the variability in clinical disease manifestations in patients with *Staphylococcus aureus* infections. *PloS one*. 2012;7:e34390.
12. Berry MP, Graham CM, McNab FW, et al. An interferon-inducible neutrophil-driven blood transcriptional signature in human tuberculosis. *Nature*. 2010;466:973-977.
13. Blohmke CJ, Darton TC, Jones C, et al. Interferon-driven alterations of the host's amino acid metabolism in the pathogenesis of typhoid fever. *J Exp Med*. 2016;213:1061-1077.
14. Mella C, Suarez-Arrabal MC, Lopez S, et al. Innate immune dysfunction is associated with enhanced disease severity in infants with severe respiratory syncytial virus bronchiolitis. *J Infect Dis*. 2013;207:564-573.
15. de Steenhuijsen Pipers WA, Heinonen S, Hasrat R, et al. Nasopharyngeal Microbiota, Host Transcriptome, and Disease Severity in Children with Respiratory Syncytial Virus Infection. *Am J Respir Crit Care Med*. 2016;194:1104-1115.
16. Ioannidis I, McNally B, Willette M, et al. Plasticity and virus specificity of the airway epithelial cell immune response during respiratory virus infection. *J Virol*. 2012;86:5422-5436.
17. Mahajan P, Kuppermann N, Mejias A, et al. Association of RNA Biosignatures With Bacterial Infections in Febrile Infants Aged 60 Days or Younger. *JAMA*. 2016;316:846-857.
18. Ramilo O, Allman W, Chung W, et al. Gene expression patterns in blood leukocytes discriminate patients with acute infections. *Blood*. 2007;109:2066-2077.
19. Anderson ST, Kaforou M, Brent AJ, et al. Diagnosis of childhood tuberculosis and host RNA expression in Africa. *N Engl J Med*. 2014;370:1712-1723.

20. Wright VJ, Herberg JA, Kaforou M, et al. Diagnosis of Kawasaki Disease Using a Minimal Whole-Blood Gene Expression Signature. *JAMA pediatrics*. 2018;172:e182293.
21. Herberg JA, Kaforou M, Wright VJ, et al. Diagnostic Test Accuracy of a 2-Transcript Host RNA Signature for Discriminating Bacterial vs Viral Infection in Febrile Children. *JAMA*. 2016;316:835-845.
22. Ardura MI, Banchereau R, Mejias A, et al. Enhanced monocyte response and decreased central memory T cells in children with invasive *Staphylococcus aureus* infections. *PLoS one*. 2009;4:e5446.
23. Allantaz F, Chaussabel D, Stichweh D, et al. Blood leukocyte microarrays to diagnose systemic onset juvenile idiopathic arthritis and follow the response to IL-1 blockade. *J Exp Med*. 2007.
24. Sweeney TE, Azad TD, Donato M, et al. Unsupervised Analysis of Transcriptomics in Bacterial Sepsis Across Multiple Datasets Reveals Three Robust Clusters. *Critical care medicine*. 2018;46:915-925.
25. Sweeney TE, Perumal TM, Henao R, et al. A community approach to mortality prediction in sepsis via gene expression analysis. *Nat Commun*. 2018;9:694.

Reviewers' comments:

Reviewer #2 (Remarks to the Author):

Summary: Overall this is an interesting study design of comparing whole blood transcriptional signatures in healthy, asymptomatic and symptomatic cCMV-infected infants with potentially important clinical implications for predicting late on-set hearing loss and intriguing insight into the biological/immunological similarities between "symptomatic" and "asymptomatic" cCMV infection. I think that the author's discussion as to the limited biological rationale for differentiating between symptomatic and asymptomatic cCMV infections is well-supported. I appreciate the authors responsiveness to reviewer's suggests and substantially improved manuscript. There are some additional clarifications that would be useful and limitations to discuss.

Major points:

1) Per line 76, infants were tested for cCMV infection in the first 21 days of life not at birth. Thus, the authors should address this clarification as to if there could be some perinatal cCMV included in their cases. This is an important caveat to the interpretations of the study design examining in utero transmission specifically and should be addressed as a limitation in the discussion

2) Additional information in the methods about how the modular analysis was completed and briefly summarize them in the results (line 227, not just the references) is necessary to make a meaningful interpretation of the biological significance of figure 3, for instance how many "modules" are measured for each category of immune response?

3) Table 1 indicates that the gestational age is significantly lower in the healthy matched controls (38 weeks) versus the cCMV infected groups (39 weeks), which should be pointed out in the results and addressed in the discussion. Does controlling for gestational age change any of the differences observed between the healthy and cCMV infected biosignatures? Could gestational age be behind the different transcriptional profiles observed in the healthy vs. cCMV-infected groups? Please address further

4) Figure 3 – please clarify the differences in the various modules (M1.2, M3.4 etc.) listed for each pathway and expand upon how the modular analysis was performed, what databases etc. are these modules based off of?

5) In the Venn diagram (supplementary figure 3), it is shown that there are overlapping transcripts between the cCMV asymptomatic and cCMV symptomatic biosignatures but not between infants in those two groups (the 0's), can you please explain this discrepancy? Why would differences in the biosignatures generated from those groups not be recapitulated?

Minor points:

6) on line 190 and throughout the manuscript they refer to the validation cohort as "independent" but this is not a formal independent replication cohort due to the shared controls, please do not refer to this validation cohort as independent given the shared healthy controls

7) line 311 should say "to infect" not "of infecting"

8) line 389 says "host genomic responses," which is awkward and unclear since it's really transcriptional responses that were assessed

9) Figure 1 shows the heatmaps for the training sets but not the validation sets, please provide both (in the supplementary materials is fine)

10) Figure 2 – legend title says fails but should say fail

11) Figure 4 – since the healthy group is used as the reference to calculate the MDTH score then it is unclear why you would perform any statistical testing comparing the cCMV groups to the control/healthy group? Please improve the legend in figure 4 to provide the clarity

Reviewer #3 (Remarks to the Author):

I thank the authors for doing a thorough job in addressing/clarifying my comments. I have no further comments to make.

REVIEWER #1

There are three points that need further attention:

There is a previous paper on expression profiles in neonatal whole blood on filter paper in children with cCMV and a healthy control group. This should be included in this manuscript.

Comment #1: The authors adequately cite and address this prior paper by Rovito et. al. (2018) in the revised introduction and discussion sections. Importantly, they highlight genes that were overexpressed in the cCMV+ infants in this group, though they did not reach statistical significance (likely due to smaller sample size compared to the current manuscript). The authors could also mention key differences in methodology since the Rovito paper examined transcriptomics from dried blood spots, which may be easier to obtain and leverage since these are collected in a standardized manner for newborn screening. Alternatively, an advance of using the whole blood for transcriptomic signatures is that there is likely less degradation of mRNA and higher sensitivity in this approach compared to the analysis on dried bloodspots. These are some of the key differences the authors could highlight further.

Response #1: We appreciate the reviewers comment as above. We now mention the advantages and disadvantages of each methodology as suggested. It reads as (page 17, lines 376-379)"

"On one hand, dry blood spots offer the advantage of leveraging samples routinely obtained in a standardized manner as part of the newborn screening, however, mRNA degradation is less likely to occur in blood samples collected prospectively for transcriptome analyses, offering higher sensitivity"

The follow-up samples are very interesting, also in view of the longer term pathogenesis. However, it is not clear to me, whether the expression profiles at the different time points were compared to the neonatal healthy control group, or to follow-up samples of the healthy control group at similar time points. This may impact the results and conclusion.

Comment #2: Though the authors do clarify how the original biosignature identified at birth/the neonatal period (first 3 weeks) is applied to the longitudinal samples in the revised manuscript, there are still concerns with how this is explained for the reader. Improved discussion of the limitation of comparing gene expression signatures in the cCMV-infected controls at 1, 2 and 3 years of age to healthy neonates NOT healthy age matched controls is a limitation for interpreting the longitudinal data and impacts the authors stated conclusions (Figure 5). The authors can simply address this by acknowledging that comparing the gene expression signatures of the cCMV-infected children at older ages to neonates is a caveat to the results since age-matched controls would be ideal.

Response #2: We appreciate the above comment and have added further clarification as recommended, specifically adding this as a limitation of the study. It reads as (page 17, lines 390-394):

Similarly, for the longitudinal analyses, the same healthy young infant controls were used for the three follow-up time points. Although analyses of those time points did not include age-matched controls, our approach allowed the initial transcriptional profiling (early infancy) to serve as a reference value over time, and suggests that the biosignature of cCMV is one of a chronic infection"

The authors should acknowledge in the discussion that their cohort reflects a more severely affected cohort than you would expect to find through universal screening. This is important because it may influence the applicability of e.g. the classifier-gene set.

Comment #3: The authors adequately address this concern in the discussion section by noting the increased severity of infection in their cohort and thus the concerns about more broad generalizability for future applications of their gene-classification signature.

Response #3: Thank you for the above comment

REVIEWER #2

Summary: Overall this is an interesting study design of comparing whole blood transcriptional signatures in healthy, asymptomatic and symptomatic cCMV-infected infants with potentially important clinical implications for predicting late on-set hearing loss and intriguing insight into the biological/immunological similarities between “symptomatic” and “asymptomatic” cCMV infection. I think that the author’s discussion as to the limited biological rationale for differentiating between symptomatic and asymptomatic cCMV infections is well-supported. I appreciate the authors responsiveness to reviewer’s suggests and substantially improved manuscript. There are some additional clarifications that would be useful and limitations to discuss.

Major points:

Comment #1: Per line 76, infants were tested for cCMV infection in the first 21 days of life not at birth. Thus, the authors should address this clarification as to if there could be some perinatal cCMV included in their cases. This is an important caveat to the interpretations of the study design examining in utero transmission specifically and should be addressed as a limitation in the discussion.

Response #1: Thanks for bringing up this comment. We used the standard definition of congenital CMV that has been utilized in several observational and multicenter interventional studies, and includes the identification of CMV (by PCR or viral culture) in the neonatal period and within the first 21 days of age¹⁻⁷. In our study the diagnosis of congenital CMV was made the first week of life and samples obtained on the first study visit at day 17 of life. We have clarified this aspect in the results section. It reads as (page 7, lines 161-163):

“The diagnosis of infants with symptomatic and asymptomatic cCMV infection was established in the first week of life by PCR, and blood samples were obtained at study enrollment the second or third week of life (median age 17 days).”

Comment #2: Additional information in the methods about how the modular analysis was completed and briefly summarize them in the results (line 227, not just the references) is necessary to make a meaningful interpretation of the biological significance of figure 3, for instance how many “modules” are measured for each category of immune response?

Response #2: Briefly, modular analysis is a systems scale strategy for transcriptome analysis that aims to reduce the abundance of transcriptional data into functional pathways or modules. The algorithm was constructed based on transcripts that were coordinately expressed across different disease states. Thus, the grouped transcripts are not disease specific, but functionally specific, allowing interpretation of the microarray data into biologically meaningful information. These coordinately expressed transcripts were grouped into modules (M), where modular over- and under-expression is defined by the percentage of transcripts within each module that are differentially expressed in the condition of interest (in this case cCMV) compared with healthy controls. A detailed description of this mining analysis strategy has been reported elsewhere⁸⁻¹⁰.

For this study of the 62 modules analyzed, 16 of them pertaining to innate and adaptive immune responses were represented in Fig 3. Of those three were related to interferon (M1.2, M3.4, M5.12), one to monocytes (M4.14), one to neutrophils (M5.15), six to inflammation (M3.2, M4.2, M4.6, 4.13, M5.1, M5.7), two T-cell related, one related to cytotoxic/NK cells (M 3.6), one plasma cell (M4.11) and one B-cell related (M4.10). The annotations of modular maps and the number of modules included in the major immune categories are summarized above and also visualized in supplementary Fig S4, S5 and S6. In addition, we have further clarified in the methods section (page 6, lines 129-134), in supplementary material, in the results section (page 11, lines 238-241) and in Figure 3 legend how modular analyses were originally derived and their interpretation.

Methods: “Briefly, modular analysis is a systems scale strategy that aims to reduce the abundance of transcriptional data into functional pathways. This approach uses clusters of coexpressed genes (or modules) to generate disease-specific transcriptional fingerprints, providing a stable framework for the visualization

and functional interpretation of gene expression data. A description of this mining analysis strategy is included in the supplementary material and has been reported elsewhere⁸⁻¹⁰

Supplementary material: “This is a systems scale and a reductionist strategy for microarray analysis in which transcriptional data is summarized as transcriptional modules. Modules are groups of genes coordinately expressed allowing functional interpretation of the microarray data into biologically meaningful information. The advantage of this approach is that is data driven, rather than derived from curated knowledge databases. Briefly, after assembling data from 12 curated transcriptome datasets generated from 410 patients with different infectious, autoimmune and immune mediated diseases (S. aureus, liver transplant, SLE, influenza, melanoma, etc) a set of 62 clusters of genes (or transcriptional modules) were identified and annotated. Modular maps were generated using a step-wise approach and visualized in a grid format, where the first round of modules (M1) is represented by the sub-network with the most genes that are co-clustered in all input datasets. In the next rounds of selection, the level of stringency to identify the core networks is relaxed, so modules are formed by genes that co-cluster in all but one of the data sets (M2), two of all the datasets (M3) and so on. Once modules are formed, they are functionally characterized as it is assumed that the co-clustering observed is driven by biological factors. For visualization purposes the significant abundance of transcripts relative to a baseline (or healthy controls) are represented by a colored dot. When the proportion of over expressed transcripts in a given module is increased, the module is represented by a red dot, while an increased proportion of underexpressed transcripts is represented by a blue dot, with the intensity of the color indicating the proportion of transcripts expressed in any given module. A detailed account of this module-based mining analysis strategy has been reported elsewhere³⁻⁷”.

Results: “Each module (M) consists of coordinately expressed genes that share a similar biologic function.^{8,11} Of the 62 modules analyzed, 16 related to innate and adaptive immune responses are represented in Fig 3 and Supplementary Fig S4, S5 and S6”.

Fig 3 legend: “Sixteen selected modules pertaining to innate and adaptive immune responses are represented above and included three related to interferon, one to monocytes, one to neutrophils, six to inflammation, two T-cell related, one related to cytotoxic/NK cells, one plasma cell, and one B-cell related”.

Comment #3: Table 1 indicates that the gestational age is significantly lower in the healthy matched controls (38 weeks) versus the cCMV infected groups (39 weeks), which should be pointed out in the results and addressed in the discussion. Does controlling for gestational age change any of the differences observed between the healthy and cCMV infected biosignatures? Could gestational age be behind the different transcriptional profiles observed in the healthy vs. cCMV-infected groups? Please address further.

Response #3: Thank you for the comment. Indeed, gestational age in the CMV infected groups (that included both the training and test sets) was 39 weeks compared to 38 weeks in healthy controls. This data was included in Table 1, as previously requested, to provide a global overview of the demographic information of study patients. Data analyses however, were performed differently and followed the standard tenets of high dimensional biological data and analysis plan. As such differential expression analysis was performed first in the derivation set (training set) to generate gene lists for the purpose of obtaining biological insight and understanding. The data was then validated on the test set of patients. When we compared the demographic characteristics of healthy controls versus symptomatic and asymptomatic cCMV infants in the training set, no significant differences were found in any of the parameters analyzed including gestational age (Table S3). Thus, gestational age was not a factor likely influencing the results of the gene expression profiles analyses. We have made a clarification under the results section (page 7, lines 163-166). It reads as:

“The overall median gestational age was 39 weeks in both cCMV groups and 38 weeks in the healthy control cohort, but there were no significant differences in gestational age between the discovery cCMV cohorts and healthy controls (Supplementary Table S3)”.

Comment #4: Figure 3 – please clarify the differences in the various modules (M1.2, M3.4 etc.) listed for each pathway and expand upon how the modular analysis was performed, what databases etc. are these modules based off of?

Response #4: Please see the response to question #2 above.

Comment #5: In the Venn diagram (supplementary figure 3), it is shown that there are overlapping transcripts between the cCMV asymptomatic and cCMV symptomatic biosignatures but not between infants in those two groups (the 0's), can you please explain this discrepancy? Why would differences in the biosignatures generated from those groups not be recapitulated?

Response #5: We apologize for the lack of clarity. The Venn diagram (Fig S3) summarized the comparisons between infants with symptomatic cCMV infection vs healthy controls (symptomatic cCMV biosignature; 2,592 transcripts), those with asymptomatic cCMV infection vs healthy controls (asymptomatic cCMV biosignature; 3,324 transcripts), and the direct comparisons of infants with symptomatic and asymptomatic cCMV included in the training sets, that yielded no significant differentially expressed genes between groups. For clarity, we have removed the latter comparison to resolve this confusion.

Minor points:

Comment #6: on line 190 and throughout the manuscript they refer to the validation cohort as “independent” but this is not a formal independent replication cohort due to the shared controls, please do not refer to this validation cohort as independent given the shared healthy controls

Response #6: We appreciate the above comment. The term independent was meant to be related to the study population (i.e. infants with congenital CMV infection), though we do appreciate the confusion this may cause as we are using the same set of controls. We have amended the manuscript as requested to provide clarity. It reads as (page 9, lines 201-202):

“...in an independent cohort of 24 symptomatic CMV-infected infants and the same healthy controls used in the derivation cohort (validation cohort; Fig 1B)”.

Comment #7: line 311 should say “to infect” not “of infecting”

Response #7: We agree with the comment and have made the above correction as suggested (page 14, line 322)

Comment #8: line 389 says “host genomic responses,” which is awkward and unclear since it’s really transcriptional responses that were assessed

Response #8: We have amended the manuscript as recommended to maintain consistency. It reads as (page 18, lines 407-409):

“Despite differences in clinical, laboratory, and neuroimaging findings, asymptomatic and symptomatic cCMV-infected infants demonstrated similar host transcriptional immune profiles”.

Comment #9: Figure 1 shows the heatmaps for the training sets but not the validation sets, please provide both (in the supplementary materials is fine)

Response #9: In this version of the manuscript, we performed principle component analysis as our method of validation as opposed to unsupervised clustering analysis that was presented in the original submission. This was done largely in response to the statistical review to ensure reproducibility of the biosignatures using a different method of validation. As requested, we have included as supplementary material the heatmaps for the validation sets (new Supplementary Figure S2)

Comment #10: Figure 2 – legend title says fails but should say fail

Response #10: We have made the change as recommended.

Comment #11: Figure 4 – since the healthy group is used as the reference to calculate the MDTH score then it is unclear why you would perform any statistical testing comparing the cCMV groups to the control/healthy group? Please improve the legend in figure 4 to provide the clarity

Response #11: For the MDTH analyses, global gene expression was compared to that of healthy controls, that served as a baseline (reference value for comparison). To derive the MDTH scores all analytical groups were included in the analyses (symptomatic cCMV, asymptomatic cCMV and controls). We used Kruskal-Wallis with Dunn’s post hoc tests for multiple comparisons. This has been highlighted and clarified in the figure legend.

REVIEWER #3:

Comment #1: I thank the authors for doing a thorough job in addressing/clarifying my comments. I have no further comments to make.

Response #1: We appreciate the thorough review and comments to further develop the manuscript.

BIBLIOGRAPHY

- 1 Kimberlin, D. W. *et al.* Effect of ganciclovir therapy on hearing in symptomatic congenital cytomegalovirus disease involving the central nervous system: a randomized, controlled trial. *J Pediatr* **143**, 16-25, doi:10.1016/s0022-3476(03)00192-6 (2003).
- 2 Stehel, E. K. *et al.* Newborn hearing screening and detection of congenital cytomegalovirus infection. *Pediatrics* **121**, 970-975, doi:10.1542/peds.2006-3441 (2008).
- 3 Ronchi, A. *et al.* Evaluation of clinically asymptomatic high risk infants with congenital cytomegalovirus infection. *J Perinatol* **40**, 89-96, doi:10.1038/s41372-019-0501-z (2020).
- 4 Kimberlin, D. W. *et al.* Valganciclovir for symptomatic congenital cytomegalovirus disease. *N Engl J Med* **372**, 933-943, doi:10.1056/NEJMoa1404599 (2015).
- 5 Marsico, C. *et al.* Blood Viral Load in Symptomatic Congenital Cytomegalovirus Infection. *J Infect Dis* **219**, 1398-1406, doi:10.1093/infdis/jiy695 (2019).
- 6 de Vries, J. J. *et al.* Real-time PCR versus viral culture on urine as a gold standard in the diagnosis of congenital cytomegalovirus infection. *Journal of clinical virology : the official publication of the Pan American Society for Clinical Virology* **53**, 167-170, doi:10.1016/j.jcv.2011.11.006 (2012).
- 7 Boppana, S. B. *et al.* Saliva polymerase-chain-reaction assay for cytomegalovirus screening in newborns. *N Engl J Med* **364**, 2111-2118, doi:10.1056/NEJMoa1006561 (2011).
- 8 Chaussabel, D. *et al.* A modular analysis framework for blood genomics studies: application to systemic lupus erythematosus. *Immunity* **29**, 150-164, doi:S1074-7613(08)00283-5 [pii] 10.1016/j.immuni.2008.05.012 (2008).
- 9 Chaussabel, D. & Baldwin, N. Democratizing systems immunology with modular transcriptional repertoire analyses. *Nature Reviews Immunology* **14**, 271-280, doi:10.1038/nri3642 (2014).
- 10 Berry, M. P. *et al.* An interferon-inducible neutrophil-driven blood transcriptional signature in human tuberculosis. *Nature* **466**, 973-977, doi:nature09247 [pii] 10.1038/nature09247 (2010).
- 11 Chaussabel, D. & Baldwin, N. Democratizing systems immunology with modular transcriptional repertoire analyses. *Nature reviews. Immunology* **14**, 271-280, doi:10.1038/nri3642 (2014).
- 12 Banchereau, R. *et al.* Host immune transcriptional profiles reflect the variability in clinical disease manifestations in patients with Staphylococcus aureus infections. *PloS one* **7**, e34390, doi:10.1371/journal.pone.0034390 (2012).
- 13 Obermoser, G. *et al.* Systems scale interactive exploration reveals quantitative and qualitative differences in response to influenza and pneumococcal vaccines. *Immunity* **38**, 831-844, doi:10.1016/j.immuni.2012.12.008 (2013).

REVIEWERS' COMMENTS:

Reviewer #2 (Remarks to the Author):

The authors were very thorough in responding to reviewer comments and providing adequate supplementary information, in particular the expanded discussion of the gene module analysis and associated results.

REVIEWERS' COMMENTS:

Reviewer #2 (Remarks to the Author):

The authors were very thorough in responding to reviewer comments and providing adequate supplementary information, in particular the expanded discussion of the gene module analysis and associated results.

Response: We appreciate the reviewer's thoughtful assessment, comments, and recommendations during the revision process.